# Antigen specificity of clonally enriched CD8⁺ T cells in multiple sclerosis

Fumie Hayashi [1,7], Kristen Mittl[1,7], Ravi Dandekar[1,7], Josiah Gerdts[1], Ebtesam Hassan[1], Ryan D. Schubert [1], Lindsay Oshiro [1], Rita Loudermilk[1], Ariele Greenfield[1], Danillo G. Augusto [1,2], Gregory Havton[1], Shriya Anumarlu[1,3], Arhan Surapaneni[1,4], Akshaya Ramesh[1], Edwina Tran[1], Kanishka Koshal[1], Kerry Kizer[1], Joanna Dreux[5], Alaina K. Cagalingan[5], Florian Schustek[5], Lena Flood[5], Tamson Moore[5], Lisa L. Kirkemo[5], Isabelle J. Fisher[1], Tiffany Cooper[1], Meagan Harms[1], Refujia Gomez[1], University of California, San Francisco MS-EPIC Team*, Claire D. Clelland[1], Leah Sibener[5], Bruce A. C. Cree[1], Stephen L. Hauser [1], Jill A. Hollenbach [1,6], Marvin Gee[5], Michael R. Wilson [1], Scott S. Zamvil [1] ✉ & Joseph J. Sabatino Jr [1] ✉

CD8⁺ T cells are the dominant clonally expanded lymphocyte population in multiple sclerosis (MS) lesions but their clonal identity, function and antigen specificity are not well understood. A comprehensive single-cell RNA-sequencing and T cell receptor-sequencing analysis of the cerebrospinal fluid and blood from individuals in the MS and control cohorts revealed a subset of 23 highly expanded and activated CD8⁺ T cell clonotypes that were enriched predominantly in the cerebrospinal fluid in the MS cohort. Using unbiased and targeted antigen discovery approaches, six CD8⁺ T cell clonotypes recognizing Epstein–Barr virus (EBV) antigens and multiple novel mimotopes were identified. Although the majority of mimotopes did not elicit functional responses, three of the expanded CD8⁺ T cell receptors from patients with MS were reactive to EBV. EBV DNA and transcripts were detected in cerebrospinal fluid, including in patients with MS who had highly expanded EBV-specific CD8⁺ T cells. These findings shed vital insight into the role of CD8⁺ T cells in MS and support an important role of EBV in MS immunopathology.

Multiple sclerosis (MS) is an inflammatory demyelinating condition of the central nervous system (CNS) characterized by substantial T cell involvement[1]. Both CD4⁺ and CD8⁺ T cells are found within the perivascular spaces as well as in the parenchyma of MS lesions[2,3]. CD8⁺ T cells are enriched and clonally expanded relative to CD4⁺ T cells in MS lesions[4,5], suggesting local antigen-driven expansion. Specific major histocompatibility complex (MHC) I alleles also alter MS susceptibility[6], providing additional support for a critical role

of cytotoxic CD8⁺ T cells in MS biology. The goal of this study was to identify T cells, particularly CD8⁺ T cells, that are uniquely expanded in the CNS and determine their phenotypic characteristics and antigen specificity.

Acquisition of CNS tissue to analyze the infiltrating T cell response in MS is invasive and rarely performed early in the disease course. Cerebrospinal fluid (CSF) is a transit site of lymphocytes entering the CNS[7] and provides a window into the immune responses within the

[1]UCSF Weill Institute for Neurosciences, Department of Neurology, University of California San Francisco, San Francisco, CA, USA. [2]Department of Biological Sciences, The University of North Carolina at Charlotte, Charlotte, NC, USA. [3]University of California Berkeley, Berkeley, CA, USA. [4]University of California Los Angeles, Los Angeles, CA, USA. [5]3T Biosciences, South San Francisco, CA, USA. [6]Department of Epidemiology and Biostatistics, University of California San Francisco, San Francisco, CA, USA. [7]These authors contributed equally: Fumie Hayashi, Kristen Mittl, Ravi Dandekar. *A list of authors and their affiliations appears at the end of the paper. ✉e-mail: zamvil@ucsf.neuroimmunol.org; joseph.sabatinojr@ucsf.edu

**Table 1 | Overview of participant characteristics**

| Cohort | n | Gender (female/male) | Mean age (yr) | Untreated (%) | OCB⁺ (%) | Active MRI (%) |
|--------|---|----------------------|---------------|---------------|----------|----------------|
| RR-MS | 11 | 7/4 | 41 | 100 | 82 | 64 |
| CIS | 2 | 2/0 | 44 | 100 | 0 | 0 |
| OND | 2 | 2/0 | 35 | 50 | 100 | Not tested |
| HC | 3 | 1/2 | 34 | N/A | 0 | Not tested |

Active MRI refers to contrast enhancement on MRI within 30 days of blood and CSF analysis; N/A, not applicable; OCB⁺, positive oligoclonal band status; RR-MS, relapsing-remitting MS.

CNS. Studies of the CSF repertoire have indeed indicated high clonal overlap with expanded T cell populations in MS lesions[8,9].

Single-cell-sequencing technologies provide powerful opportunities for deep phenotyping and clonal characterization of T cells in numerous diseases, including MS[10–14], yet detailed analyses of CSF-expanded T cells and their antigen specificity in MS are lacking. In this study, CSF and blood from individuals with early untreated MS as well as control participants were subjected to single-cell RNA-sequencing (scRNA-seq) paired with single-cell T cell receptor (TCR)-sequencing (scTCR-seq). After identifying a subset of CSF-expanded CD8⁺ T cells in the patients with MS, their antigen specificity was investigated using a combination of unbiased and targeted antigen discovery methodologies.

## Results

### Study participants

A total of 18 individuals were enrolled in the study—11 with relapsing-remitting MS, two with clinically isolated syndrome (CIS), two with other neuro-inflammatory disorders (OND) and three healthy controls (HCs). The demographics of the four cohorts are presented in Table 1 and those of each individual in Supplementary Table 1. All of the patients in the MS and CIS (MS/CIS) cohort were treatment-naive (that is, no previous history of immunomodulatory or immunosuppressive therapies) at the time of sample collection but one of the patients with OND was on immunotherapy with a TNF-α inhibitor.

### Identification of T cell subsets by single-cell sequencing

Paired peripheral blood and CSF were collected on the same day for each study participant. Freshly acquired samples comprised of all unseparated cell subsets underwent paired scRNA-seq and scTCR-seq using 10X Genomics 5′ library preparation kits to permit combined single-cell transcriptional phenotyping and TCR clonal analysis. The scRNA-seq data of all participants in this study were previously published[15]. All major immune cell subsets were readily identified from the scRNA-seq data, with T cell clusters comprising the largest fractions (Fig. 1a). To characterize conventional TCRαβ T cells, all subsequent analyses focused on only those T cells with paired scRNA-seq and scTCR-seq data (Fig. 1b). A total of 48,468 individual T cells were identified from the blood and CSF across all participants (Fig. 1c). As expected, TCR-associated genes (CD3E and CD3D) were highly upregulated with minimal expression of non-T cell-associated genes (for example, CD19; Extended Data Fig. 1). We identified 33,349 CD4⁺ and 15,119 CD8⁺ T cells expressing paired TCRαβ genes (described in Methods) for analysis from the combination of blood and CSF of all 18 participants (Supplementary Table 2).

Pseudotime analysis of T cells in the CSF revealed distinct populations of T cells largely segregated based on T cell subsets (that is, CD4⁺ or CD8⁺), highlighting the distinct transcriptional signatures associated with different T cell states and functions (Fig. 1d). Both CD8⁺ and CD4⁺ T cells were distinct between the peripheral blood and CSF (Fig. 1e,f). For instance, CD8⁺ T cells (Supplementary Table 3) in the CSF displayed significantly increased expression of various genes relative to the peripheral blood, including genes associated with migration

and trafficking (CXCR3, CXCR4, CCL4, ITGB1 and ITGA4), signaling and activation (CD2, FYN and DUSP2), and cytotoxicity (GZMK and GZMA). In contrast, peripheral blood-derived CD8⁺ T cells expressed significantly higher levels of FOS, JUN, DUSP1 and GADD45B, indicating an alternate activation state. In a comparison of only memory (CD27-expressing) CD8⁺ T cells, there was significant upregulation of genes associated with T cell activation (HLA-DRA), chemokines (CCL4 and XCL1) and cholesterol metabolism (LDLR and SQLE), and downregulation of genes associated with T cell signaling (FOS, FOSB, JUN and JUNB) in the CSF relative to the blood (Extended Data Fig. 2a and Supplementary Table 4). In the CSF, CD4⁺ T cells (Supplementary Table 3) had significantly increased expression of genes similar to their CD8⁺ T cell counterparts (ITGB1, ITGA4, CXCR3, GZMA, GZMK and CD2) as well as distinct genes (JUN, FOS, DUSP1, CCR7 and HCST).

Given the disproportionate number of participants in the different disease categories (Table 1), we grouped the patients with MS or CIS (MS/CIS; n = 13) and performed differential gene expression against the combined group of HCs and patients with OND (HC/OND; non-MS group; n = 5; Fig. 1g). In CD8⁺ T cells combined from the peripheral blood and CSF, various genes were differentially expressed between the MS/CIS and HC/OND groups (Fig. 1h). In particular, genes associated with tissue trafficking (CXCR4, CCL5, KLF2, ITGA4, ITGB1 and CD69) and cytotoxicity (GZMK, KLRG1 and GZMA) were upregulated in the MS/CIS cohort (Supplementary Table 5). In contrast, genes associated with central memory status (CCR7, SELL and LEF1) and TCR signaling (CD8B, CD3D, CD3E, LCK, ZAP70 and LAT) were downregulated relative to the HC/OND group. A similar profile was observed for CD4⁺ T cells from the blood and CSF of patients with MS/CIS, including increased expression of genes related to tissue migration (ITGB1, CD69, ITGA4, CXCR3 and CXCR4) and cytokine secretion (IL32 and GZMK), and reduced central memory status (CCR7, LEF1, SELL and TCF7) and TCR signaling (LCK; Fig. 1i and Supplementary Table 5). Overall, these data suggest that both CD8⁺ and CD4⁺ T cells in the patients with MS/CIS are more activated with increased effector functions and tissue homing capacity than the HC/OND cohort, consistent with other studies[11,12].

### T cell clonal analysis

The clonal repertoire of T cell subsets was compared across compartments (that is, peripheral blood versus CSF) and across disease states (that is, MS/CIS versus HC/OND). T cell clonotypes were defined as T cells sharing identical V and J genes and CDR3 amino-acid sequences for paired TCRαβ sequences similar to previous studies[10,16]. A total of 31,756 unique CD4⁺ T cell clonotypes and 10,825 unique CD8⁺ T cell clonotypes were identified from all individuals (Supplementary Table 2). The CD4⁺ and CD8⁺ T cell diversities (measured by Shannon entropy) were significantly higher in MS/CIS compared with HC/OND in the peripheral blood and CSF (Extended Data Fig. 3). The diversity of CD4⁺ T cells was significantly higher in blood compared with CSF but not for CD8⁺ T cells (Extended Data Fig. 3). These findings suggest a more diverse array of CD4⁺ and CD8⁺ T cell clonotypes are present in both the blood and CSF of patients with MS/CIS.

T cell clonal expansion is a hallmark of previous antigen encounter; therefore, our analysis focused on T cell clonality in the CSF. T cells were divided into three different categories of clonal expansion: unexpanded (single cell of a given clonotype), moderately expanded (>1 cell but <0.75% of the CSF T cell repertoire of an individual) or highly expanded (≥0.75% of the CSF T cell repertoire of an individual). We chose 0.75% as the cutoff for highly expanded clonotypes as it represented a relatively high threshold based on the clonal expansion observed in the CSF of our patient cohorts as well as others[10,17]. Although small fractions of clonally expanded CD4⁺ T cells were observed in the CSF, much larger populations of highly and moderately expanded CD8⁺ T cells were observed in similar proportions in the MS/CIS and HC/OND groups (Fig. 2a). CD4⁺ and CD8⁺ T cell clonal expansion in the CSF was overall similar between the patients with CIS or MS (Supplementary Table 6).

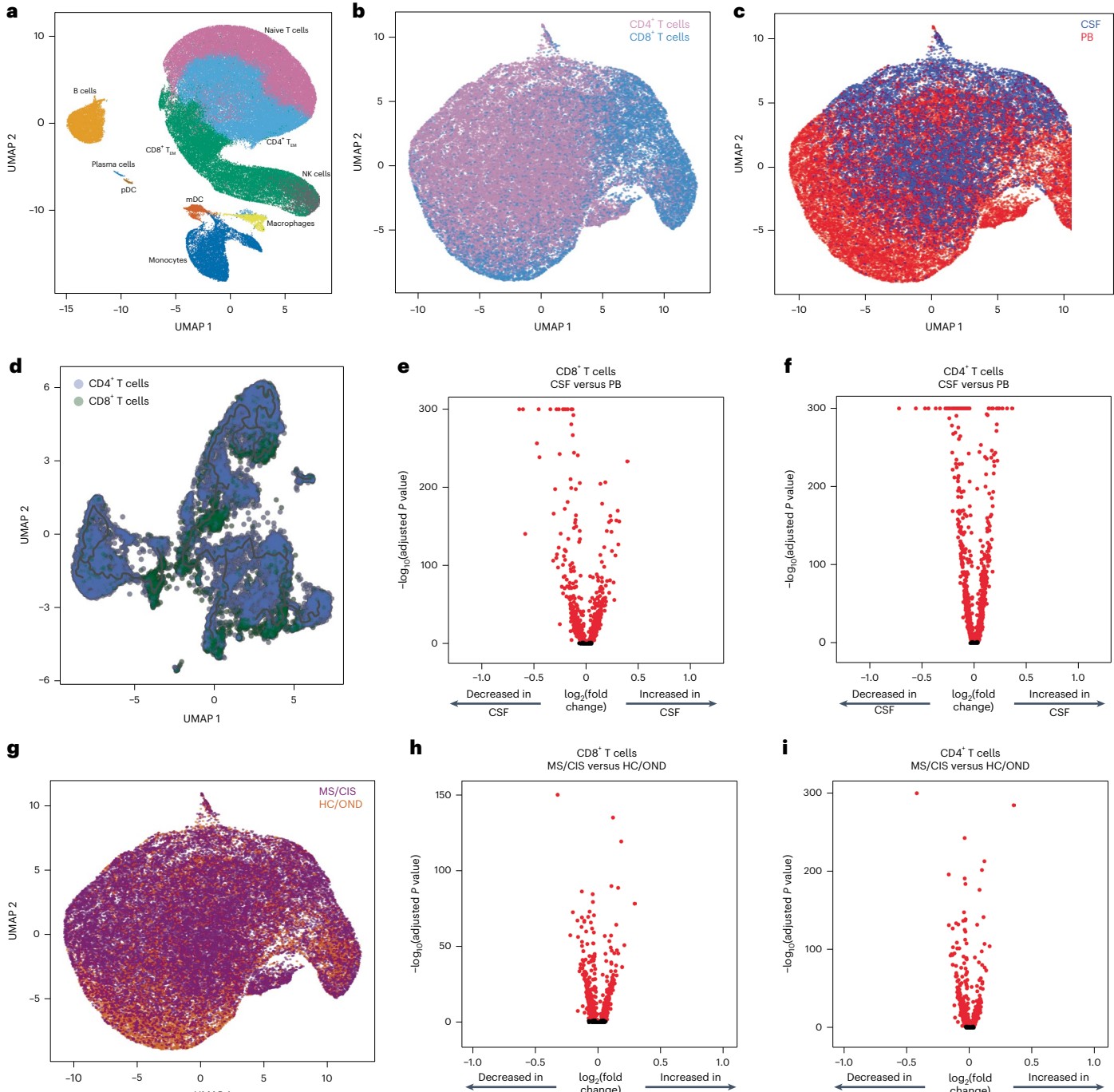

**Fig. 1 | Single-cell-sequencing analysis of T cells in blood and cerebrospinal fluid. a**, Major immune cell subsets from combined blood and CSF of all patients were identified by scRNA-seq. mDC, myeloid dendritic cells; NK, natural killer cells; pDC, plasmacytoid dendritic cells; $T_{EM}$, effector memory cells. **b,c,g**, T cells were defined after integration of the scRNA-seq and scTCR-seq data, allowing segregation of T cells by CD4/CD8 status (**b**), compartment (CSF; **c**) and disease status (**g**). **d**, Pseudotime trajectory analysis of CD4+ and CD8+ T cells in CSF and peripheral blood (PB). **e,f,h**, Analysis of differential gene expression between CSF-derived and PB-derived CD8+ (**e**) and CD4+ (**f**) T cells as well as between MS/CIS-derived and HC/OND-derived CD8+ (**h**) and CD4+ (**i**) T cells. Differential gene expression comparisons were performed using a two-sided Wilcoxon ranked-sum test with Bonferroni correction (adjusted *P*). Genes with adjusted *P* < 0.05 are indicated in red. UMAP, uniform manifold approximation and projection.

## Cerebrospinal fluid enrichment of highly expanded T cell clonotypes

To delineate between T cells expanded similarly in the blood and CSF versus those preferentially expanded in the CSF, the abundance of all T cell clonotypes in the blood and CSF was compared in all individuals. The overwhelming majority of T cell clonotypes were detected in the blood or CSF only, whereas only about 1.5% of all clonotypes were found in both compartments (Fig. 2b). We postulated that highly expanded T cell clonotypes (that is, CSF frequency of ≥0.75%) that were enriched in the CSF relative to the peripheral blood were more likely to be responsive to local antigens in the CSF and/or CNS (albeit not necessarily CNS-specific antigens). Enriched CSF-expanded T cell clonotypes were defined as those with a CSF frequency at least twofold higher than the peripheral-blood frequency from the same individual. This yielded 33 highly CSF-enriched and expanded T cell clonotype varying from approximately twofold to more than 100-fold higher frequencies in the

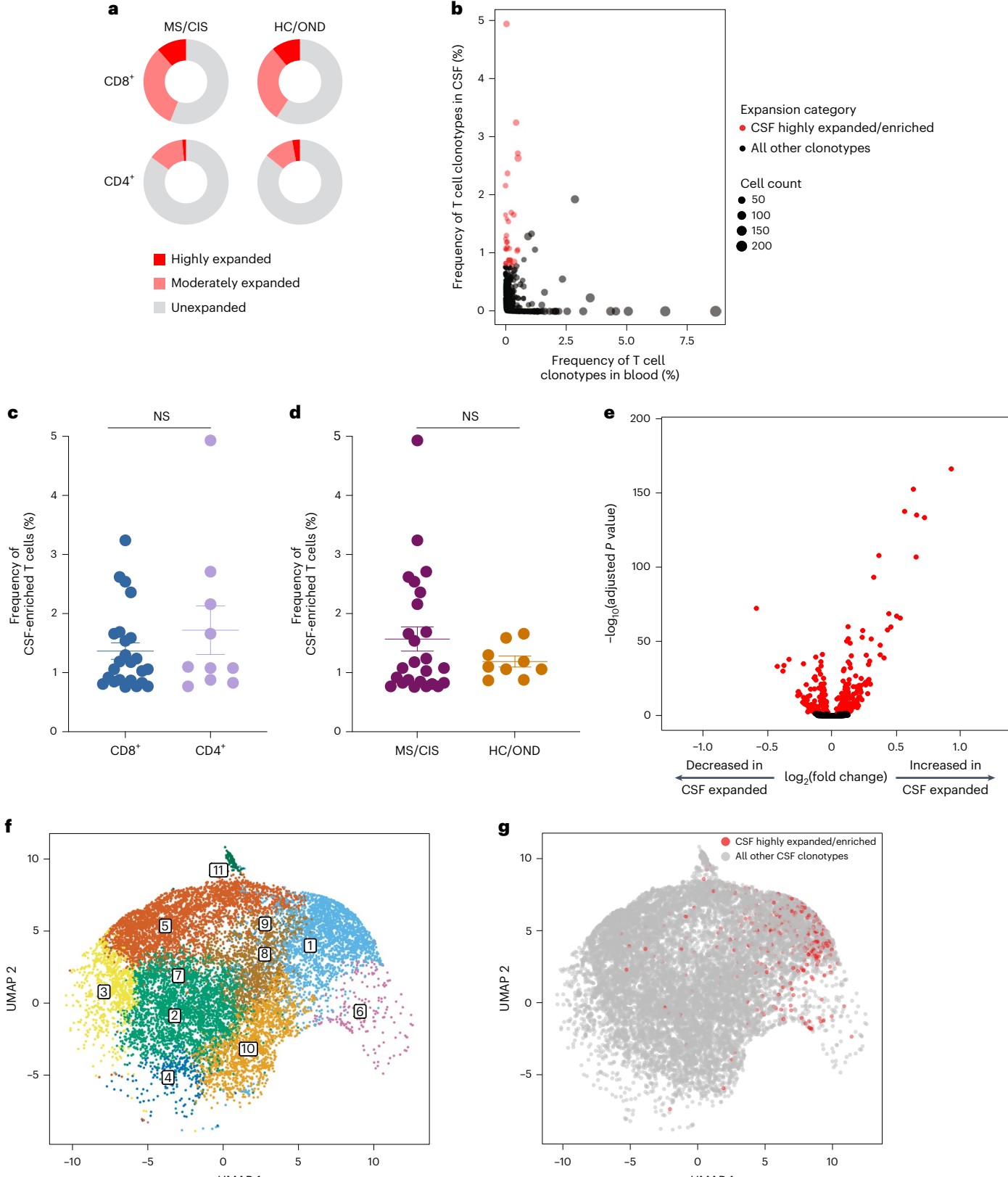

**Fig. 2 | T cell clonal expansion in cerebrospinal fluid. a**, CD8[+] and CD4[+] T cell clonal expansion was compared between MS/CIS and HC/OND subjects. **b**, Clonal frequencies of all T cell clonotypes in the CSF and blood that were highly expanded T cells and enriched at least twofold more frequently than the blood of the same individual are highlighted in red. **c**,**d**, Frequency of highly expanded and enriched T cells according to CD8 (n = 24) or CD4 (n = 10) status (**c**) and MS/CIS (n = 25) or HC/OND (n = 9) status (**d**). Data are the mean ± s.e.m.; unpaired two-tailed Student's *t*-test with Welch's correction; NS, not significant. **e**, Analysis of differential gene expression between highly expanded and unexpanded T cells in the CSF. Two-sided Wilcon ranked-sum test with Bonferroni correction; genes with adjusted *P* < 0.05 are indicated in red. **f**,**g**, Unbiased clustering of all CSF T cells (**f**; the 11 distinct clusters are numbered) overlaid with highly expanded/enriched T cells (**g**).

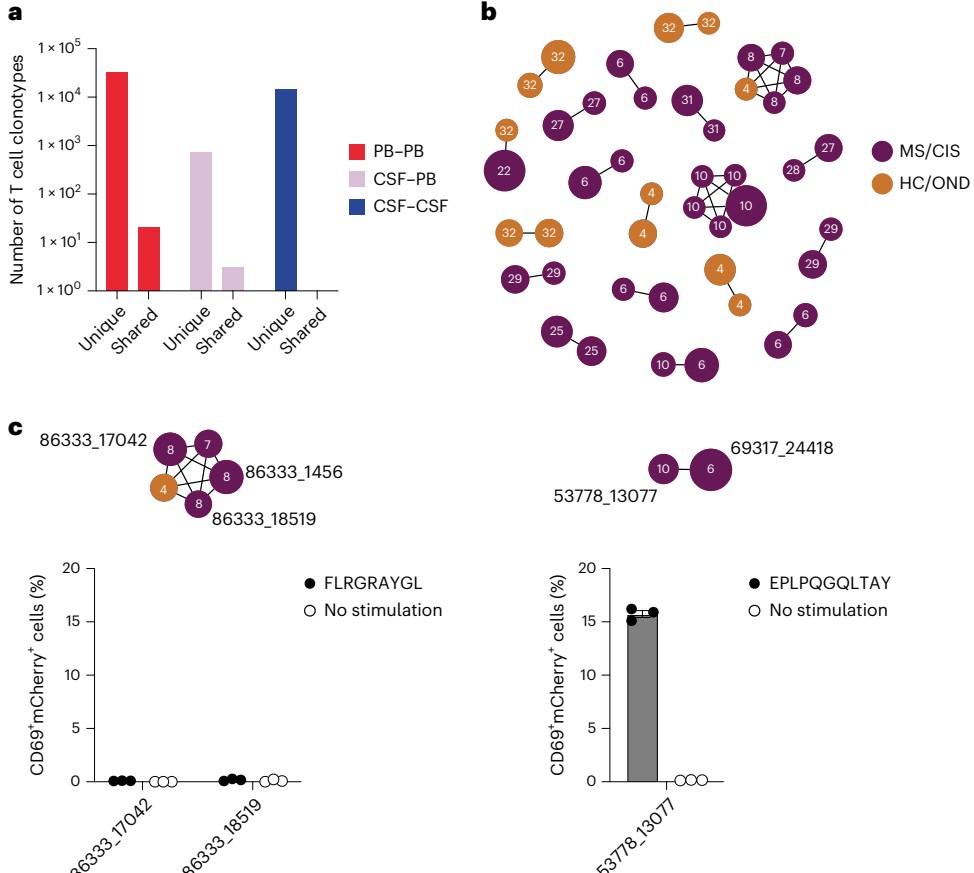

**Fig. 3 | T cell clonal relationships. a**, Number of unique or shared T cell clonotypes between different compartments from all study participants. PB, peripheral blood. **b**, GLIPH2 analysis of highly expanded and enriched T cell clonotypes in the CSF compared with all other CSF T cell clonotypes. Clonal size is indicated by node size and clonally related populations are connected by lines. The number in each clonotype refers to the participant ID. **c**, The indicated GLIPH2-aligned CD8⁺ TCRs to the EBV-specific TCRs 86333_1456 (left) and 69317_24418 (right) were expressed in reporter Jurkat cells and tested for reactivity to the corresponding EBV peptides ($n$ = 3) or no-stimulation control ($n$ = 3). Data are the mean ± s.e.m. FLRGRAYGL (EBV EBNA3A$_{193–201}$) was presented by HLA-B*08:01-expressing APCs (left) and EPLPQGQLTAY (EBV BZLF1$_{54–64}$) was presented by HLA-B*35:01-expressing APCs (right).

CSF relative to peripheral blood (Fig. 2b and Supplementary Table 7). More than 70% of the highly expanded and CSF-enriched T cell clonotypes in the CSF were CD8⁺ T cells. The frequencies of highly expanded CSF-enriched CD8⁺ and CD4⁺ T cells were similar, ranging from 0.76 to 4.9% of the entire CSF repertoire of an individual (Fig. 2c). Although there were no statistically significant differences in the mean frequencies of highly expanded CSF-enriched T cell clonotypes between MS/CIS and HC/OND, only participants in the MS/CIS cohort had CSF-enriched T cells with frequencies greater than 2% (Fig. 2d). One patient with MS (patient identifier (ID), MS6) had 11 highly enriched T cell clonotypes, the majority of which were CD8⁺ T cells, which encompassed nearly 20% of their CSF repertoire (Supplementary Table 7). These findings therefore provide strong support for robust oligoclonal CD8⁺ T cell expansion and enrichment in the CSF, with the greatest expansion found in MS/CIS.

**Single-cell transcriptomics of cerebrospinal fluid-expanded T cells**

Highly expanded and unexpanded T cells in the CSF were compared by scRNA-seq analysis. Substantial differential gene expression changes were observed in highly expanded T cells in comparison to their unexpanded counterparts (Fig. 2e and Supplementary Table 8). In particular, genes associated with cytotoxic CD8⁺ T cell function (*CD8A*, *CD8B*, *NKG7*, *KLRD1*, *GZMA*, *GZMH*, *GZMM*, *GZMK* and *EOMES*) and chemotaxis (*CCL5* and *CCL4*) were significantly increased in

highly expanded T cells, whereas genes associated with naive status were significantly reduced (*IL7R*, *LTB* and *LDHB*). Targeted gene expression analysis revealed increased expression of additional genes associated with effector/memory differentiation (*KLRG1* and *CD27*), tissue homing (*CXCR3* and *CCR5*) and resident memory status (*CD69* and *IGTAE*) as well as inhibitory genes associated with chronic antigen exposure (*HOPX*, *TIGIT*, *DUSP2*, *PDCD1* and *LAG3*) in highly CSF-expanded CD8⁺ and CD4⁺ T cells (Extended Data Fig. 4). A tissue-resident-memory (T$_{RM}$) phenotype of CSF-expanded T cells coexpressing *CD69* and *IGTAE* was confirmed by the reduced expression of *KLF2* and *S1PR1* genes (Extended Data Fig. 2b,c and Supplementary Table 9). In contrast, CSF-unexpanded T cells expressed higher levels of genes associated with central memory/ nonactivation (*SELL*, *CCR7*, *IL7R*, *TCF7* and *LEF1*) as well as the integrin gene *ITGB1*. To further characterize CSF-enriched and expanded T cell clonotypes, the 33 T cell clonotypes were overlaid with 11 distinct CSF T cell clusters (Fig. 2f,g). The overwhelming majority of the enriched and expanded clonotypes were found in cluster 1, which was defined by a significantly increased expression of a number of genes associated with cytotoxic effector CD8⁺ T cells, including *CD8A*, *CD8B*, *PLEK*, *DUSP2*, *EOMES*, *GZMK*, *GZMA*, *GZMH*, *PRF1*, *NKG7*, *CCL5* and *CCL4* (Supplementary Table 10). Overall, these data indicate that highly clonally expanded T cells in the CSF express gene profiles indicative of substantial antigen experience, cytotoxicity and distinct tissue homing capacities.

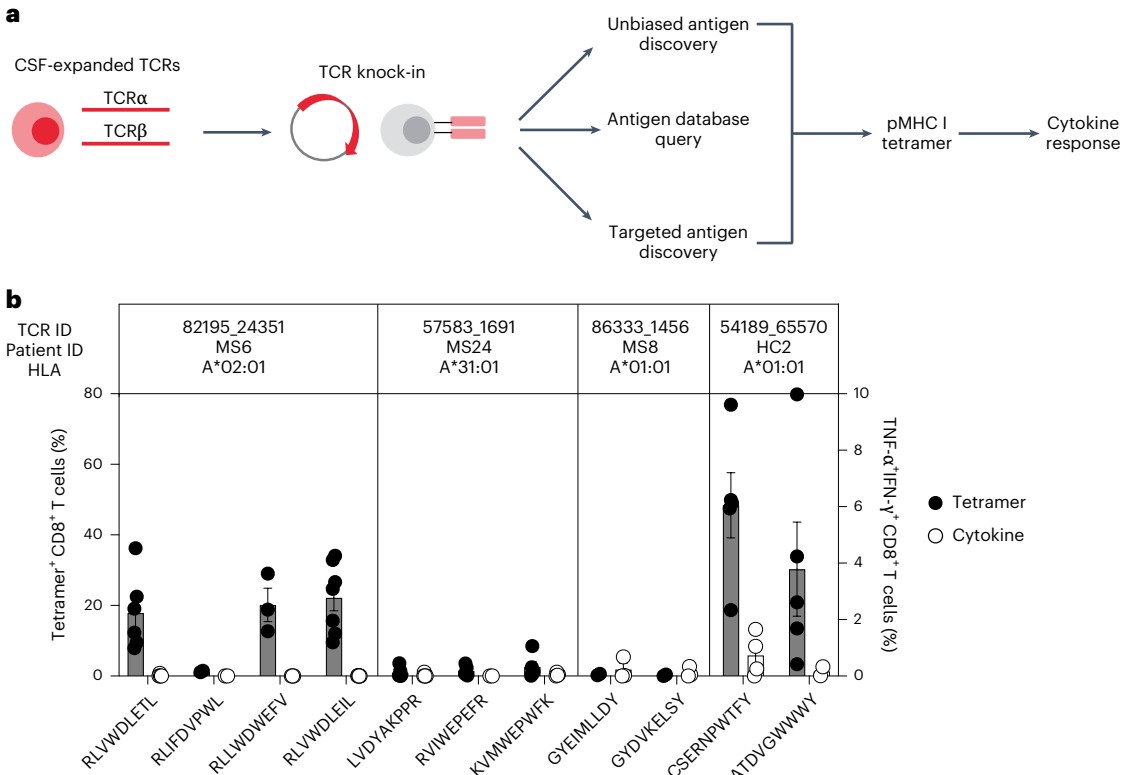

**Fig. 4 | Antigen discovery of highly expanded cerebrospinal fluid-enriched CD8+ T cells. a**, Individual TCRαβ pairs were cloned into plasmids and expressed in primary human CD8+ T cells by nonviral CRISPR knock-in. Candidate antigens for testing specificity were identified in three parallel strategies, screened by pMHC tetramer binding and validated by cytokine production to cognate antigen. **b**, Candidate antigens for four TCRs identified by pMHC yeast display (unbiased antigen discovery) were tested for tetramer binding and cytokine reactivity experiments. Data are the mean ± s.e.m. Each peptide was tested a minimum of two times using T cells from different donors for tetramer and cytokine, respectively (RLVWDLETL, $n = 6$ and 4; RLIFDVPWL, $n = 2$ (both tests); RLLWDWEFV, $n = 3$ (both tests); RLVWDLEIL, $n = 7$ and 6; LVDYAKPPR, $n = 6$ and 5; RVIWEPEFR, $n = 5$ and 2; KVMWEPWFK $n = 5$ and 4; GYEIMLLDY $n = 2$ and 3; GYDVKELSY $n = 2$ and 3; CSERNPWTF $n = 5$ and 4; ATDVGWWWY, $n = 5$ and 2).

## Clonal relationships of expanded cerebrospinal fluid-enriched T cells

Nearly all CSF T cell clonotypes across all individuals were unique. Only 21 identical TCRs (that is, same V and J genes and CDR3 amino-acid sequences for the paired α and β chains) were found between the peripheral blood of different individuals and another three that were identical between the blood and CSF of different individuals, irrespective of disease status (Fig. 3a). To further assess clonal relationships, Grouping of Lymphocyte Interactions with Paratope Hotspots 2 (GLIPH2) was employed, an algorithm to help identify TCRs with potentially shared specificity based on sequence similarity within the CDR3β region[18]. All CSF T cell clonotypes were analyzed using GLIPH2 and the output was then queried against the 33 CSF high-enriched CDR3 sequences. Using this approach, 19 clonally related networks comprised of a total of 44 clonotypes were identified (Fig. 3b and Supplementary Table 11). Most of the networks comprised two related clonotypes and two networks were comprised of five clonotypes each. Almost all networks consisted of clonotypes from the same individual and were identified primarily among the individuals with MS or CIS (Fig. 3b). Nearly all of the clonally related T cells were CD8+ T cells (Supplementary Table 11), suggesting potential shared antigen specificity.

## Antigen discovery of highly expanded cerebrospinal fluid-enriched CD8+ T cells

Antigen discovery efforts focused on the 23 highly expanded, CSF-enriched CD8+ T cell clonotypes (≥0.75 of the CSF repertoire and enriched at least twofold in the CSF relative to the blood) that comprised more than 70% of the expanded CSF-enriched T cells. Several different strategies were undertaken (Fig. 4a). An unbiased antigen discovery approach was first employed using a peptide:MHC (pMHC) yeast display library in which approximately $1–10 × 10^8$ random peptides are displayed on a given MHC allele for probing recognition against individual TCRs[19]. Of the 23 CD8+ TCRs, 18 were successfully expressed and tested against specific MHC I allele libraries based on library availability and the alleles of the participants from whom the TCRs were derived. Four TCRs (three MS/CIS and one HC) demonstrated substantial enrichment of specific peptides from three different MHC I libraries (Supplementary Table 12).

To validate these candidate antigens, each TCR was expressed individually in primary human CD8+ T cells by nonviral CRISPR–Cas9-mediated TCR knock-in (Extended Data Fig. 5a). Candidate TCR-expressing CD8+ T cells were then probed for antigen specificity using pMHC tetramers loaded with peptides identified from the yeast display library screen. Three of the four tested TCRs demonstrated robust tetramer binding to most or all of the library-identified peptides (Fig. 4b and Extended Data Fig. 5b). The ability of CD8+ T cells expressing these TCRs to respond functionally to the same antigens was tested by intracellular cytokine stimulation using antigen-presenting cells (APCs) expressing the relevant MHC I allele. Strikingly, only TCR clonotype 54189_65570 demonstrated cytokine production to peptide CSERNPWTFY; none of the other TCR-expressing CD8+ T cells were functionally responsive to the respective yeast display-derived peptides (Fig. 4b). As nearly all of the yeast display peptides identified by pMHC I tetramers were not naturally occurring (that is, mimotopes), the analysis was extended to an array of foreign and human peptide homologs (Supplementary Table 12). Varying degrees of tetramer

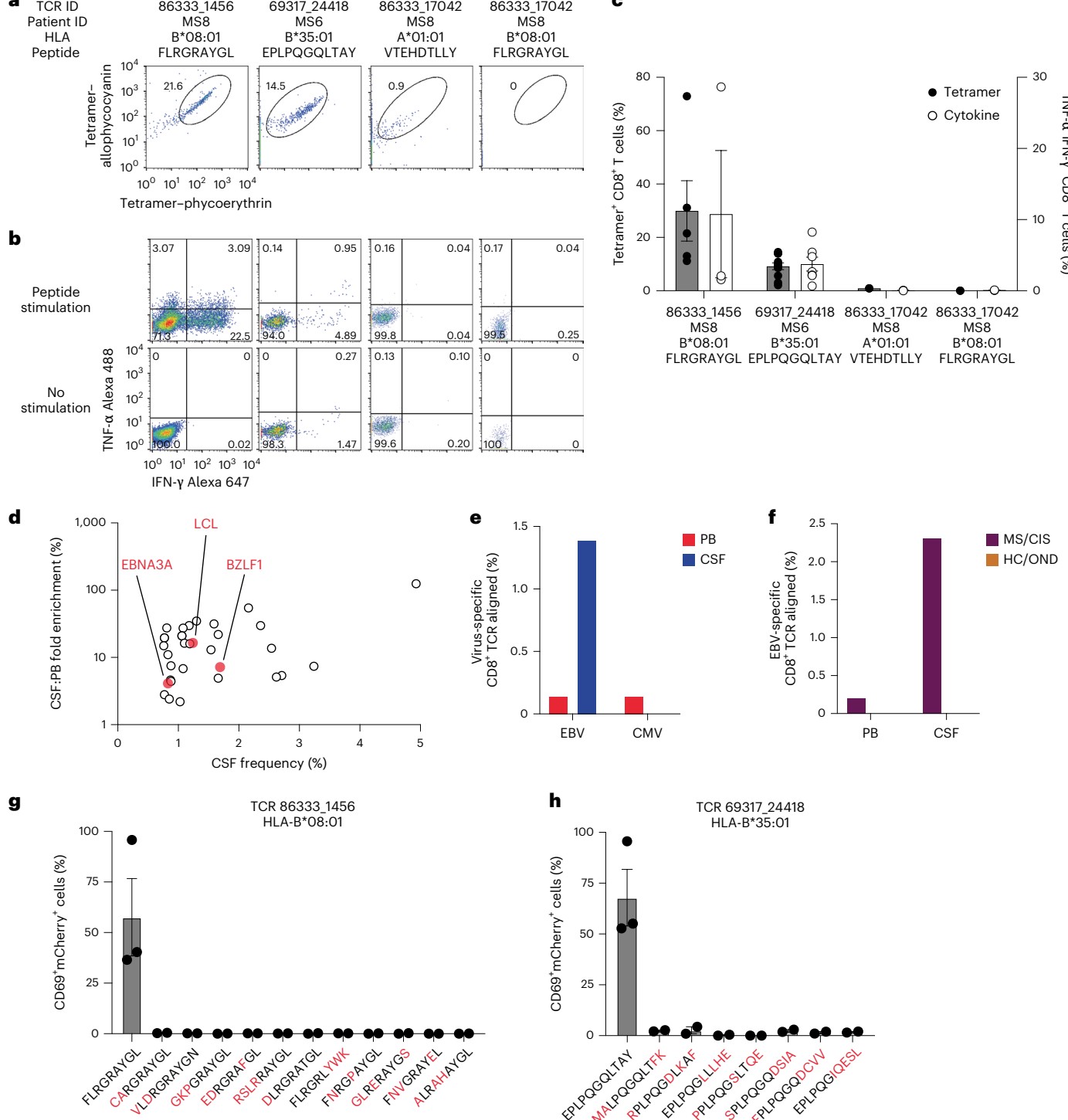

**Fig. 5 | EBV reactivity of highly expanded cerebrospinal fluid-enriched CD8⁺ T cells. a,b**, Representative flow cytometry analysis of tetramer binding (**a**) and cytokine production (**b**) of three TCRs (from patients with MS) with predicted reactivity to four different viral epitopes. The percentage of cells in the gated regions are indicated. **c**, Percentage tetramer binding and cytokine reactivity of each TCR. Cytokine reactivity reflects the subtracted background from the no-stimulation control. Data are the mean ± s.e.m.; TCR 86333_1456 FLRGRAYGL, $n = 5$ tetramer and 3 cytokine; TCR 69317_24418 EPLPQGQLTAY, $n = 11$ tetramer and 9 cytokine; TCR 86333_17042 VTEHDTLLY and TCR 86333_17042 FLRGRAYGL, $n = 2$ (all groups). **d**, Frequencies and degree of enrichment of the three EBV-specific clonotypes (highlighted in red) relative to all other highly enriched and expanded

T cell clonotypes. **e**, TCR sequencing alignment of expanded CD8⁺ TCRs to EBV- and CMV-specific TCRs in the peripheral blood (PB) and CSF. **f**, EBV-specific TCR alignment of all expanded CD8⁺ T cell clonotypes in the PB and CSF of MS/CIS and HC/OND study participants. **g,h**, Summary of functional reactivity of Jurkat cells expressing the indicated TCR specific for EBV EBNA3A$_{193-201}$:B*08:01 (**g**) or EBV BZLF1$_{54-64}$:B*35:01 (**h**) to the indicated peptides ($n = 3$ EBV peptides and 2 peptide homologs). Responses reflect the mean ± s.e.m. frequency of CD69 and NFAT–mCherry double-positive cells with the no-stimulation background control subtracted. Amino-acid differences between cognate EBV peptides (leftmost of each plot) and self-peptide homologs are indicated in red. Each peptide was tested in a minimum of two independent experiments.

binding were observed depending on the TCR tested but none of the peptide homologs elicited cytokine responses above background (Extended Data Fig. 6a,b). Thus, although the unbiased antigen discovery approach yielded novel mimotopes of several CD8+ T cell clonotypes detectable by pMHC I tetramer binding, none exhibited functional reactivity to naturally occurring antigens.

### Probing viral specificity of clonally expanded CD8+ T cells

The highly expanded CSF-enriched TCR clonotypes were queried against several public TCR databases, including VDJdb[20], TCRex[21] and TCRmatch[22], as an additional TCR antigen discovery strategy (Fig. 4a). One CD8+ T cell clonotype (86333_1456) from participant MS8 demonstrated an exact match to both TCRα and -β sequences with a well-described Epstein–Barr virus (EBV) epitope, EBNA3A$_{193-201}$ (FLRGRAYGL; Supplementary Table 13), which is restricted by HLA-B*08:01, an allele carried by this individual (Supplementary Table 1). This identical clonotype was also moderately expanded in the CSF from an individual with Alzheimer's disease[17]. A second CD8+ T cell clonotype (69317_24418) from participant MS6 was a near-exact match to a TCR specific for the EBV epitope BZLF1$_{54-64}$ (EPLPQGQLTAY) restricted by HLA-B*35:01, an allele also carried by this individual.

These TCRs were expressed in primary human CD8+ T cells as described earlier and their specificity was again tested by pMHC I tetramer analysis. TCR 86333_1456 and TCR 69317_24418 showed robust tetramer staining to EBNA3A$_{193-201}$:B*08:01 and BZLF1$_{54-64}$:B*35:01, respectively (Fig. 5a). To confirm functional reactivity, primary human CD8+ T cells expressing each of these TCRs were stimulated with APC lines expressing cognate HLA and loaded with or without cognate EBV peptide. Each TCR demonstrated clear cytokine production to the relevant EBV peptide (Fig. 5b,c), confirming both CSF-expanded and enriched CD8+ T cell clonotypes are specific to EBV antigens.

In light of these findings, the possibility that additional CSF-enriched and expanded CD8+ T cells may be specific for viral antigens, in particular EBV, was further explored. Note that severe acute respiratory syndrome coronavirus 2 peptides were not tested as all samples were collected previous to the coronavirus disease 2019 pandemic. Nineteen TCRs were tested against panels of pMHC I tetramers loaded with previously identified immunodominant viral epitopes restricted by HLA matching that of the TCR donors. A total of 98 peptides restricted by eight different MHC I alleles were screened (Supplementary Table 14). Each TCR was screened with individual pMHC tetramers, except in the case of HLA-A*02:01 where tetramers were pooled in groups of five due to the large number of candidate peptides. Each TCR was tested against the indicated peptides a minimum of two times using two different T cell donors for TCR expression. No specific tetramer signal was observed for any of the TCRs to any of the peptides beyond the two EBV epitopes already identified for TCRs 86333_1456 and 69317_24418 (Extended Data Fig. 7). Although TCR 86333_17042 from participant MS8 showed an identical match for a TCRβ sequence specific for EBV and cytomegalovirus (CMV) antigens with corresponding MHC I alleles (Supplementary Table 13), it did not show any notable tetramer binding or cytokine reactivity to either viral antigen (Fig. 5a–c).

Potential reactivity to EBV was further assessed for the other CSF-expanded and enriched CD8+ T cells given the EBV reactivity of two clonotypes. EBV-transformed lymphoblastoid cell lines (LCLs) were employed to survey a wide array of processed EBV epitopes across a multitude of HLA alleles. NFAT–mCherry-expressing Jurkat cells transfected with the CD8 co-receptor and a single candidate TCR were co-cultured with partially HLA-matched LCLs. Fully HLA-mismatched LCLs and TCR-expressing Jurkat cells from HLA-mismatched patients were used as negative controls. This enabled testing of 16 additional candidate TCRs against at least two different LCLs matching 3–6 MHC I alleles (Supplementary Table 15). Almost all TCRs showed no detectable reactivity; however, TCR 94669_8198 from patient MS27 demonstrated a clear reproducible response to LCLs only when matching the

HLA-A*29:02 allele (Fig. 6a–c). No response was elicited from primary B cells from the same donor used to generate the LCLs, indicating this is very likely to be an EBV-specific response rather than a B cell self-antigen or alloreactive response. To identify a potential specific EBV epitope, Jurkat reporter cells expressing TCR 94669_8198 were tested against seven candidate EBV peptides identified from The Immune Epitope Database (FLYALALLL, VFGQQAYFY, AYSSWMYSY, FVYGGSKTSLY, VFSDGRVAC, VSSDGRVAC and ILLARLFLY) presented by HLA-A*29:02-expressing APCs. No functional response was elicited, however, indicating reactivity to a still unspecified EBV epitope (Extended Data Fig. 8).

We also explored EBV specificity for CD8+ T cell clonotypes that were aligned to the highly CSF-expanded and enriched CD8+ T cell clonotypes that were EBV-reactive (Fig. 3b). Only GLIPH2-derived TCR sequences from CD8+ T cells that shared the same MHC I allele as that of the aligned EBV-specific clonotype were tested (Supplementary Table 16). Unlike the EBV-specific clonotype 86333_1456, the GLIPH2-aligned TCRs 86333_17042 and 86333_18519 (all from MS8) showed no detectable reactivity to EBNA3A peptide FLRGRAYGL restricted by HLA-B*08:01 (Fig. 3c). TCR 53778_13077 from MS10 was aligned by GLIPH2 to TCR 69317_24418 from MS6, specific for the BZLF1 B*35:01-restricted peptide EPLPQGQLTAY. Strikingly, TCR 53778_13077 was found by VDJdb search to exactly match a TCR previously demonstrated to be specific for EPLPQGQLTAY[23]. This specificity was validated by stimulating Jurkat reporter cells expressing TCR 53778_13077 with or without EPLPQGQLTAY presented by HLA-B*35:01-expressing APCs. Notably, TCR 53778_13077 was moderately expanded in the CSF (0.35%) and enriched approximately three-fold relative to the blood of MS10 (Supplementary Table 2).

These findings indicate that at least three highly expanded CD8+ T cells in the CSF of patients with MS are specific for EBV, but the specificities for most of the enriched T cell clonotypes remain unknown (Fig. 5d). The overwhelming majority of clonotypes were not enriched in the CSF of the 18 study participants (Fig. 2b and Supplementary Table 2) or the three patients with MS and EBV-specific clonotypes (Extended Data Fig. 9a). To assess whether EBV specificity among expanded CD8+ T cells in the CSF was overall enhanced compared with the blood, TCR sequencing alignment (identical V genes, J genes and CDR3 amino-acid sequences for paired TCRα and -β chains) was performed against all expanded CD8+ TCR sequences (>1 TCR per clonotype) in VDJdb[20] with CMV used as a comparison. EBV- and CMV-aligned CD8+ TCR sequences in the blood were very similar; however, EBV specificity was markedly increased in the CSF, whereas no CMV specificity was found (Fig. 5e). When the EBV-aligned CD8+ TCR sequences of the MS/CIS and HC/OND groups were compared, EBV-specific TCR alignment was only observed in the patients with MS/CIS (Fig. 5f). This provides additional support that EBV-specific CD8+ T cell expansion is uniquely increased in the CSF in MS.

### Transcriptional profiles of EBV-specific CD8+ T cells

CD8+ T cells of different viral specificities can exhibit distinct phenotypic characteristics[24]. The transcriptional profiles of the three CSF-expanded CD8+ T cell clonotypes were therefore compared against all other CSF-expanded and enriched CD8+ T cells. Differential gene expression analysis revealed three genes that were significantly increased in the EBV-specific CD8+ T cells, most notably *CXCR5* (Supplementary Table 17), which is associated with migration to B cell follicles and control of chronic infections[25]. Specific genes associated with memory differentiation, migration and tissue residency were also compared. Consistent with previous reports[24], CD27 was particularly abundant in the expanded EBV-specific CD8+ T cells (Extended Data Fig. 9b). In addition, *KLF2*, *CXCR4*, *S1PR1* and *CCL4* were more abundant in EBV-specific CD8+ T cells. In contrast, *CXCR3*, *CD69* and *CCL5* were more abundant in non-EBV-specific expanded CD8+ T cells. These findings therefore indicate a distinct phenotype of CSF-expanded CD8+ T cells that are specific for EBV. Rather than

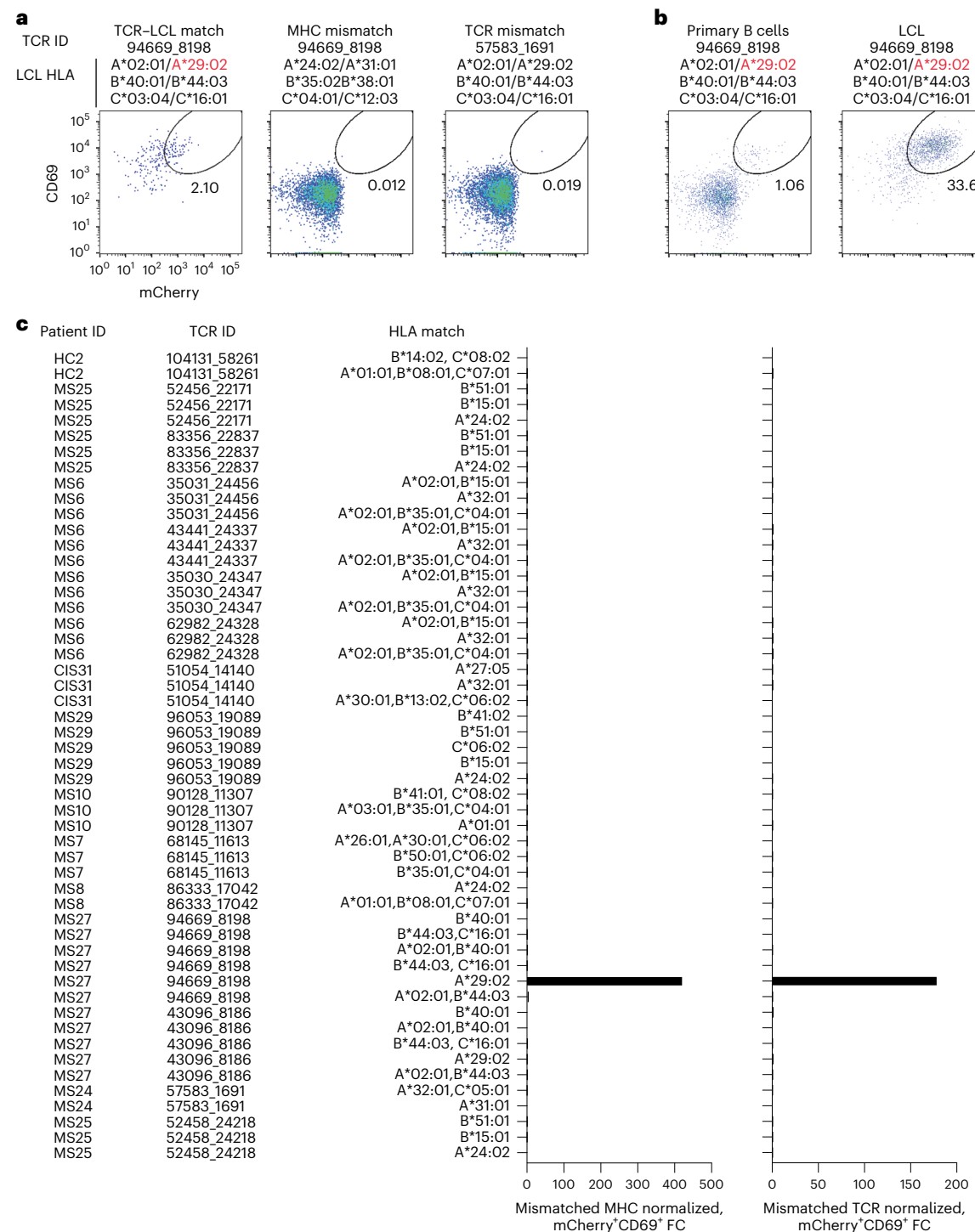

**Fig. 6 | Cerebrospinal fluid-expanded CD8⁺ T cell reactivity to EBV-transformed B cells. a**, Representative flow cytometry analysis of Jurkat reporter cells expressing the indicated TCR and CD8 co-receptor that were co-cultured with partially HLA-matched (matching allele indicated in red) EBV-transformed LCLs. Reactivity was assessed by coexpression of CD69 and NFAT–mCherry. Fully HLA-mismatched LCLs and mismatched TCR-expressing Jurkat cells were used as negative controls. **a,b**, The percentage of cells in the gated regions is indicated. **b**, Representative flow cytometry analysis of Jurkat

reporter cells expressing TCR 94669_8198 were co-cultured with LCLs or primary uninfected B cells from the same donor. **c**, Summary of all candidate TCRs tested and the corresponding matching MHC I alleles expressed by different LCL lines. The mCherry⁺CD69⁺ signal of a given TCR-expressing Jurkat cell line co-cultured with partially MHC-matched LCLs was normalized to the signal observed from completely MHC-mismatched LCLs (left) or mismatched TCR-expressing Jurkat cells (right), which was reported as fold change (FC).

expressing T$_{RM}$ markers and genes associated with lymphocyte recruitment, these findings suggest that EBV-expanded CD8⁺ T cells in the CSF are an effector population associated with follicular homing and B cell interactions.

## Lack of self-antigen cross-reactivity of EBV-specific CD8⁺ T cells
To determine whether the two EBV peptide-reactive CD8⁺ T cell clonotypes may be cross-reactive against self-antigens, the TCRs were screened against panels of self-peptides with partial sequence

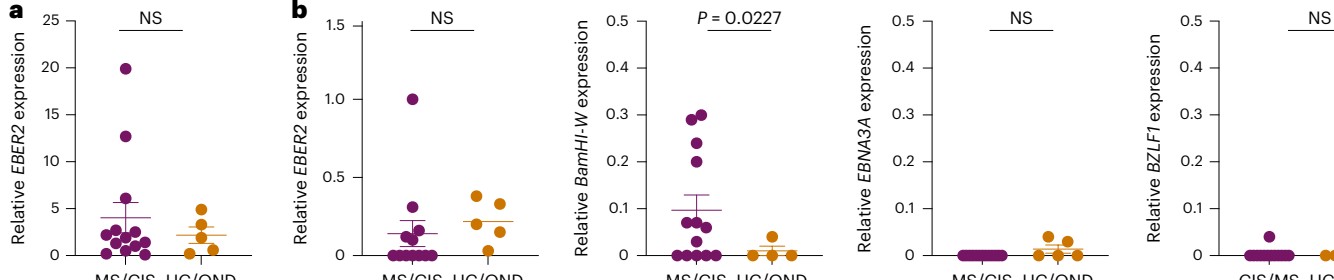

**Fig. 7 | Detection of EBV DNA and transcripts in cerebrospinal fluid. a,** Summary of EBV DNA ddPCR results from CSF supernatant in which EBER2 was normalized to a housekeeping gene (MS/CIS, *n* = 13; HC/OND, *n* = 5). **b,** EBV cDNA for each of the indicated genes were measured by ddPCR and normalized to a housekeeping gene. Each sample was run in duplicate and each dot represents the average result from each study participant. Data are the mean ± s.e.m. MS/CIS and HC/OND samples were compared using an unpaired two-tailed Student's *t*-test with Welch's correction; NS, not significant; *n* = 13 for MS/CIS for all genes except *EBER2* where *n* = 12 due to lack of sufficient sample for MS27 and *n* = 5 in HC/OND for all genes except *BamHI-W* where *n* = 4 due to a lack of sufficient sample for OND4.

homology (Supplementary Table 18). Using NFAT–mCherry-expressing Jurkat cells transfected with the CD8 co-receptor and TCR 86333_1456 or 69317_24418, high reactivity to the respective EBV peptides was confirmed (Fig. 5g,h). Strikingly, no notable reactivity was observed for any of the self-peptide homologs. Although this does not entirely exclude the possibility for self-antigen cross-reactivity, it raises the possibility that the CSF enrichment of these clonally expanded CD8[+] T cells may be driven by reactivity to EBV.

### Presence of EBV in cerebrospinal fluid

To assess for the presence of EBV in CSF, DNA was extracted from the CSF of all study participants and PCR amplified with primers specific for the EBV *BZLF1* gene (Extended Data Fig. 10a). The PCR amplicons were Sanger sequenced for further confirmation (Supplementary Table 19). In this manner, EBV DNA was detected in 6/13 MS/CIS samples and 2/5 HC/OND samples (Supplementary Table 20), including patients MS6 and MS8 who also harbored highly expanded EBV-reactive CD8[+] T cells in their CSF. The presence of EBV DNA was further quantified by droplet digital PCR (ddPCR) via amplification of the *EBER2* gene normalized to a housekeeping gene. EBV was detected in the CSF of nearly all patients and control study participants, although the relative abundance varied widely with the highest levels found in patients with MS/CIS (Fig. 7a and Extended Data Fig. 10b). EBV transcripts to several latent and lytic genes were also assessed by complementary DNA quantification. *EBER2* cDNA was overall less detectable than DNA and there was no significant difference between MS/CIS and HC/OND (Fig. 7a,b). *EBNA3A* (latency III gene) and *BZLF1* (early lytic gene) were mostly undetectable with no significant difference between the two cohorts (Fig. 7b). Strikingly, a significant increase was observed in *BamHI-W* transcripts in the MS/CIS cohort, including patients MS6, MS8 and M27 with highly expanded EBV-specific CD8[+] T cells in CSF. This therefore suggests that EBV reactivation is enhanced in the CSF in patients with MS/CIS, which may drive the expansion of EBV-specific CD8[+] T cells.

### Discussion

CD8[+] T cells are the dominant lymphocyte population in MS lesions[2,12], where they are highly clonally expanded[4,5,8,9], suggesting reactivity to hitherto unknown local antigens. Although previous studies have explored changes in gene expression and T cell clonal expansion in the CSF of patients with MS[10–12,26], numerous questions remain regarding the identity of clonally expanded CD8[+] T cells and their antigen specificity in MS. Our comprehensive transcriptional and clonal analysis identified CSF-infiltrating T cells with increased expression of genes associated with T cell activation, the $T_{RM}$ phenotype and CNS migration in the MS/CIS cohort, consistent with previous reports[10–12,27]. As clonally expanded CD8[+] T cells are present in the CSF in normal physiologic conditions and in CNS pathology[10,17,26], identification of MS-specific CD8[+] T cell clonal

populations remains a challenge. Invoking a strategy used to identify disease-relevant T cells in inflammatory arthritis[16,28] and cancer[29], a subset of highly clonally expanded and CSF-enriched CD8[+] T cells that had the highest frequencies in the patients with MS/CIS was identified. It was noteworthy that more than 70% of the highly expanded and CSF-enriched T cell clonotypes were CD8[+] given that more than twice as many CD4[+] T cells were analyzed. These CSF-enriched T cell clonotypes were widely characterized by a highly differentiated, antigen-experienced and cytotoxic phenotype with high CNS-trafficking potential, consistent with other reports[30]. These gene signatures were very similar to *GZMB* and $T_{RM}$ markers enriched in CD8[+] T cells in MS lesions[3,31], strongly suggesting these T cell clonotypes are CNS-infiltrating.

Small networks of highly expanded CSF-enriched T cells with shared TCR sequence features to other less-expanded clonotypes were found, which overwhelmingly occurred within the same individual. These findings suggest that distinct, clonally expanded T cells may be contributory to MS pathology, unlike other autoimmune conditions with preferential TCR usage[16]. Combined with the inherent technical challenges in T cell antigen discovery, these findings highlight the difficulties in identifying the antigen specificity of clonally expanded T cells in MS.

The majority of studies on candidate T cell auto-antigens in MS have focused on CD4[+] T cells[32–34]. Through the use of three parallel antigen discovery strategies, our study provides substantial new insight into the antigen specificity of CD8[+] T cells in MS. Novel mimotopes to several MS-derived CD8[+] TCRs were identified by pMHC yeast display, a powerful unbiased antigen screening tool. Although the majority of mimotopes and naturally occurring peptide homologs were readily detectable by pMHC I tetramers, only one elicited a measurable functional response. The reason for the discrepancy between pMHC tetramer binding and functional reactivity is unclear but could be due to the absence of catch bonds by high-affinity TCR ligands[35]. Nonetheless, these candidate peptides provide an important framework for identifying TCRs with similar specificities in other individuals.

The methodology of testing individual TCRs in primary human T cells by pMHC tetramer screening, followed by validation with functional reactivity is highly rigorous and ensured only genuine positive results. This approach was particularly important in the case of a TCR that demonstrated an exact TCRβ match to another antigen-specific clonotype yet did not share the same specificity, highlighting the need to validate every TCRαβ individually. Antigen specificity should therefore be interpreted cautiously when based solely on partial TCR sequence matching.

Three distinct CSF-expanded and enriched CD8[+] T cell clonotypes specific for EBV antigens were identified from three different patients in the MS cohort. Although EBV-specific CD8[+] T cells have been previously reported in the CSF of MS and other neuro-inflammatory conditions[36–39], the present study used paired TCRαβ analysis to

unequivocally demonstrate EBV reactivity of highly enriched and clonally expanded CSF CD8[+] T cell populations in MS. These findings are particularly relevant in light of recent evidence that EBV infection is a prerequisite for the subsequent development of MS[40]. Interestingly, the EBNA3A:B*08:01-specific CD8[+] TCR identified in one of the patients with MS participating in this study was highly related to expanded CD8[+] T cell clonotypes previously found in several patients with Alzheimer's disease[17]. Our findings therefore provide further support that EBV may be related to multiple forms of CNS pathology.

The mechanism by which EBV is involved in MS pathogenesis remains unresolved. The fact that the EBV-specific CD8[+] T cell clonotypes identified here were highly expanded and enriched in the CSF suggests these T cells may be responding to antigen in the CNS. EBV-specific B cells and CD4[+] T cells in MS have been suggested to be cross-reactive to CNS autoantigens[41–43] (that is, molecular mimicry). We were unable to demonstrate cross-reactivity of the two EBV peptide-specific CD8[+] T cell clonotypes against partially homologous self-peptides, but this does not completely rule out such a mechanism. Alternatively, the findings of CD8[+] T cells reactive against EBV late latent and lytic antigens are consistent with other reports[3,36,39] and could indicate EBV reactivation in the CNS[44,45]. In addition to the detection of EBV DNA in the CSF of most study participants, the increased expression of EBV transcripts in the MS/CIS cohort suggests that EBV reactivation drives expansion of EBV-specific CD8[+] T cells. These findings are consistent with other recent results[46] and suggest that EBV-specific CD8[+] T cell expansion in the CSF could be a protective response to control reactivated EBV in MS.

There are multiple mechanisms by which EBV could gain access to the CNS. In addition to primary infection of cells within the CNS[47], a number of studies have described the induction of 'atypical' T-bet[+]CXCR3[+] B cells by EBV[48,49], which could enable their migration into the CNS. EBV expression is highly dynamic, permitting the virus to exist in various latency or lytic programs[47]. It is plausible that dysregulation of EBV expression is relevant to MS pathology. Clinical trials using adoptive T cell therapies targeting EBV in MS did not show a clear benefit in progressive MS[50], however, it remains unclear how such therapies may alter EBV viral loads and expression as well as relapse and magnetic resonance imaging (MRI) outcomes. It is also important to consider that EBV reactivation in MS may represent an epiphenomenon as memory B cell differentiation into plasma cells is a trigger of EBV reactivation. Thus, it is possible that EBV reactivation and expansion of EBV-specific CD8[+] T cells are simply markers of B cell activation, which is ultimately the driver of MS pathology independent of EBV.

This study was limited by the smaller population of control participants. Follow-up studies with larger numbers of well-matched MS and control participants are needed to more clearly identify disease-relevant T cell populations in MS. In addition, longitudinal analyses of T cell clonal expansion in earlier versus later stages of MS are needed. Although the transcriptional phenotyping analyses suggest a pro-inflammatory cytotoxic phenotype of CSF-expanded CD8[+] T cells, further in vitro and in vivo analyses are needed to determine what role these cells play in MS. It is also important to acknowledge that despite the rigor of the antigen specificity testing, this approach was not exhaustive and was limited in the breadth of antigens that were tested. Given that various foreign and self-antigens are considered viable antigenic targets in MS, future studies will need to incorporate high-throughput approaches to probe multiple target antigens simultaneously. The detection of EBV in the CSF of individuals without MS probably reflects the fact that the majority of the general population, with or without MS, is chronically infected with EBV. Given that there are only trace B cells in the CSF and CNS of healthy individuals, it is possible that cell-free EBV DNA originated in the blood. Alternatively, there are other cellular reservoirs in the CNS where EBV has been identified even in healthy individuals[47].

Elucidating the role of CD8[+] T cells in MS requires the assessment of their clonal repertoire in the CNS, identification of their antigenic targets and determination of their in vivo functions. This study provides important progress towards all three aims by demonstrating a small population of predominantly CD8[+] T cells that were highly expanded and enriched in the CSF of patients with MS with strongly upregulated genes associated with antigen exposure, CNS migration and cytotoxicity. The finding of EBV specificity of three of these CD8[+] T cell clonal populations in the presence of EBV helps to advance the understanding of MS pathogenesis and may permit the development of novel disease biomarkers and therapies.

## Online content

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

## Methods

### Research ethics statement

The studies in this Article have been approved by the University of California San Francisco (UCSF) Institutional Review Board research ethics committee (protocol numbers 10-02389 and 14-15278). Informed consent was obtained from all participants in this study. No compensation was provided to the study participants.

### Study cohort

The participants, MS/CIS and control, were enrolled through the UCSF ORIGINS or Expression, Proteomics, Imaging, Clinical (EPIC) studies (https://epicstudy.ucsf.edu/). Healthy controls and patients with OND were enrolled in the biobanking study 'Immunological Studies of Neurologic Subjects'. All enrolled participants with MS or CIS were diagnosed according to the 2017 McDonald criteria[51]. Basic demographic and clinical information for all research participants is shown in Table 1 and Supplementary Table 1.

### Single-cell library preparation

Blood and CSF samples were collected from the enrolled participants on the same day during clinical and research procedures after informed consent. CSF (20–30 ml) was collected by lumbar puncture from each individual. Blood and CSF were processed immediately after collection in preparation for single-cell library preparation as previously described[15]. Unfractionated peripheral blood mononuclear cells (PBMCs) were isolated using CPT mononuclear cell preparation tubes (BD Biosciences) and resuspended in 2% fetal bovine serum (FBS). The CSF was centrifuged at 400$g$ and 4 °C for 15 min, resuspended in 80 μl supernatant and counted. Single-cell sequencing libraries were prepared using 5′ scRNA-seq and 5′ T cell V(D)J scTCR-seq kits (10X Genomics).

### Single-cell sequencing analysis

Raw data for both scRNA-seq and scTCR-seq datasets were processed using CellRanger (v3.0.1 and v3.1.0, respectively) by 10X Genomics. The cellranger count and cellranger vdj commands were run with input Ensembl GRCh38.v93 and GRCh38.v94 references for the scRNA-seq and scTCR-seq data, respectively. All data were analyzed using a custom bioinformatics pipeline that included Seurat (v3.1.2–v4.3.0), the Spliced Transcripts Alignment to a Reference (STAR) algorithm[52] (v2.5.1), SingleR[53] (v1.1.7) and DoubletFinder[54] (v2.0.2). TCR V(D)J contig assemblies outputted from CellRanger were further annotated and analyzed using Immcantation (v3.1.0). TCR clonal families were identified using Change-O[55] (v0.4.6), which generated clone IDs for both TCRα and -β chain assemblies. The scRNA-seq data have been uploaded to the Gene Expression Omnibus (GEO) repository under BioProject PRJNA549712 (GEO accession number GSE133028) and the scTCR-seq data have been uploaded under BioProject PRJNA1232831 (GEO accession number GSE291328).

### Quality control for single-cell data

Across both RNA-seq and VDJ data, reads present in more than one sample that shared the same cell barcode and unique molecular identifier were filtered using previously described methods[15]. The R package DropletUtils was used to filter out these reads in the RNA-seq data and SingleCellVDJdecontamination (https://github.com/UCSF-Wilson-Lab/SingleCellVDJdecontamination) was used to apply the same methods to filter out these reads in the VDJ data.

All gene counts from scRNA-seq data were combined using Seurat. Only genes present in two or more cells were included. Only cells containing transcripts of between 700 and 2,500 genes were included. The PercentageFeatureSet function was used to calculate the percentage of mitochondrial transcript expression for each cell. Cells that expressed at least 10% mitochondrial genes were omitted. Gene counts were normalized using the R package SCTransform[56]. The parameter do.correct. umi was set equal to TRUE and var.to.regess was set to nCount_RNA.

All filtered cells were clustered using 20 principal components (PCs) in Seurat. Clusters were formed using a shared nearest-neighbor graph in combination with dimensional reduction using uniform manifold approximation and projection[57]. Doublet detection and removal were performed for each sample using DoubletFinder[54] with expected doublet rates set based on the 10X Genomics reference manual. Cumulative sums were iteratively calculated for each PC to measure the per cent variance accounted for with the data. To determine a reasonable number of PCs, a threshold of 90% variance was applied, which resulted in 12 PCs being inputted when reclustering cells. Clusters of cells with a high expression of platelet markers (*PPBP* and *PF4*) or hemoglobin subunits (*HBB*, *HBA1* and *HBA2*) were omitted. Among the remaining cells, all V gene transcripts (*TRAV*, *TRBV*, *IGHV*, *IGKV* and *IGLV*) were removed and an additional round of reclustering was performed with nine PCs.

Assembled TCR contigs outputted from CellRanger were inputted into the Immcantation pipeline for a second round of alignments to the VDJ region using IgBLAST. Contigs containing fewer than three unique molecular identifiers were omitted. Only contigs that aligned in frame (both the FUNCTIONAL and IN_FRAME output fields were TRUE) and across the constant region were retained. Cells in the TCR VDJ data were only kept if these contained one TCRβ chain and one TCRα chain. If cells had multiple chains, TCRα or -β, which passed these thresholds, the contig with the largest number of unique molecular identifiers and or reads was kept.

### Cell-type annotation and differential gene expression analysis

Cell-type annotations were generated using previously described methods[15]. Cell types were defined by performing differential gene expression analysis for each cluster. The normalized gene expression profile for each cluster was compared with the remaining cells using a Wilcoxon rank-sum test using the FindAllMarkers function (min. pct = 0.1, logfc.threshold = 0.25, return.thresh = 0.01). The most upregulated genes, with the highest positive average log-transformed fold change, were compared with a custom panel of canonical gene makers (Supplementary Table 21) spanning several key immune cell types—B cells, CD4+ and CD8+ T cells, natural killer (NK) cells, classical monocytes, inflammatory monocytes, macrophages, plasmacytoid dendritic cells and monocyte-derived dendritic cells. In addition to these manual cell-type annotations, another set of cell types was determined using the automated cell-type annotation tool SingleR, which used the combined Blueprint and ENCODE reference dataset for fine-tuning predictions[53].

A T cell subset was created by filtering for cells that overlap both RNA-seq and TCR VDJ data. All clusters annotated as T cells had their annotations modified by CD8 gene expression. Among the T cells, any cell with *CD8A* or *CD8B* expression was annotated as a CD8+ T cell. The remaining T cells were then annotated as CD4+ T cells. Differential gene expression analyses were performed using the FindMarkers command in Seurat with the Wilcoxon test and the following parameters: *P*-adjusted value cutoff = 0.05 and log(fold change) cutoff = 0.

### T cell immune repertoire analysis

The TCR contigs outputted from Immcantation were clustered based on similarities between their TCR variable region genes (*TRAV* or *TRBV*), TCR joining region genes (*TRAJ* or *TRBJ*) and complementary determining region 3 (CDR3) amino-acid sequences. A TCR clonotype was defined as cells containing TCRα and -β chains, each containing identical V and J genes, and CDR3 amino-acid sequences. Cell counts were computed for each clone ID, including separate cell counts for PBMC and CSF samples. Shannon's entropy was calculated between CSF and PBMC samples of different disease groups using the alphaDiversity function in the R package alakazam. Specifically, the exponential of diversity scores (*D*) from the Shannon–Wiener index were extracted from the output of alphaDiversity by filtering for the diversity order ($q = 1$). Clonal expansion was defined as clones containing more than one cell. Among the expanded clonal families of TCRs, CSF enrichment of highly expanded

clones was determined by the ratio of the CSF to peripheral-blood frequencies. Clones with a CSF frequency of ≥0.75% were annotated as CSF highly expanded. Clones that were expanded in CSF (that is, more than singletons), but with frequencies less than highly expanded, were labeled as moderately expanded. Among the expanded clonal families of TCRs, CSF enrichment of highly expanded clones was determined by the ratio of the CSF to peripheral-blood frequencies. CSF highly expanded TCRs were inputted into GLIPH2 to generate glyph groups, which indicated which TCRs were predicted to target the same epitope. These glyph groups were used to create a network using the R package igraph and graphically displayed using Cytoscape.

## HLA genotyping

HLA sequencing was performed as previously described, adapted to include HLA class II[58]. For each sample, 100 ng high-quality DNA was fragmented using a Library preparation enzymatic fragmentation kit 2.0 (Twist Bioscience). After fragmentation, the DNA was repaired, and poly(A) tails were attached and ligated to Illumina-compatible dual-index adapters with unique barcodes. After ligating, the fragments were purified with a 0.8× ratio of AMPure XP magnetic beads (Beckman Coulter). Double size selection was performed (0.42× and 0.15× ratios) and libraries of approximately 800 bp were selected, at which point libraries were amplified and purified using magnetic beads. After fluorometric quantification, each sample was pooled (30 ng per sample) via ultrasonic acoustic energy. A Twist target enrichment kit (Twist Bioscience) was then used to perform target capture on pooled samples. Sample volumes were then reduced using magnetic beads and DNA libraries were bound to 1,394 biotinylated probes. Probes were designed specifically to target all exons, introns and regulatory regions of the classical HLA loci, including HLA-A, HLA-B, HLA-C, HLA-DPB1, HLA-DRB1 and HLA-DQB1. Next, streptavidin magnetic beads were used to capture fragments targeted by the probes. The captured fragments were then amplified and purified. A BioAnalyzer instrument (Agilent) was then used to analyze the enriched libraries. After evaluation, the enriched libraries were sequenced using a paired-end 150-bp sequencing protocol on the NovaSeq platform (Illumina). Following sequencing, HLA genotypes were predicted using HLA Explorer (Omixon).

## TCR cloning

The TCR sequences for each α and β gene pair were codon-optimized and used to generate gene blocks (IDT) in which the TCRβ and TCRα sequences were separated by a P2A sequence. Flanking homology arms were included to permit knock-in into the human *TRAC* locus, as previously described[59]. The gene blocks were cloned into pUC19 plasmids by Gibson assembly and the sequence was verified by Sanger sequencing.

## Primary human T cell culture

Primary human CD8[+] T cells were isolated from commercially purchased leukopaks (Vitalant or Stemcell; unidentified healthy donors). PBMCs were isolated by Ficoll centrifugation and cryopreserved before each experiment. In all experiments, T cells were cultured in RPMI medium containing 10% FBS, 2-mercaptoethanol, penicillin–streptomycin with L-glutamine, sodium pyruvate, MEM vitamin solution and nonessential amino acids (all Fisher Scientific). TCR knock-in was performed as previously described[59], with minor changes. Briefly, CD8[+] T cells were isolated from thawed PBMCs by negative selection (Miltenyi) and rested overnight with 5 ng ml⁻¹ human IL-7. The CD8[+] T cells were stimulated 1:1 with anti-human CD3/CD28 magnetic Dyna beads (Fisher Scientific), 20 ng ml⁻¹ human IL-2, 5 ng ml⁻¹ human IL-7 and 5 ng ml⁻¹ IL-15 for 48 h previous to T cell electroporation.

## Ribonucleoprotein production for TCR knock-in

Guide RNAs specific for the human *TRAC* locus were generated by incubating CRISPR RNA (crRNA; AGAGTCTCTCAGCTGGTACA) 1:1 with *trans*-activating crRNA (Dharmacon) for 30 min at 37 °C to yield a final concentration of 80 µM. Polyglutamic acid (0.8× volume) was added to the guide RNA as previously described[60]. Cas9 (QB3; Macrolab) was added 1:1 with the guide RNA and incubated for 15 min at 37 °C to yield a 20 µM ribonucleoprotein, which was used immediately for electroporation.

## TCR knock-in of primary human T cells

Dyna beads were removed from the T cell culture using an EasySep separation magnet (StemCell) 48 h after CD8[+] T cell stimulation. The T cells were then centrifuged at 200*g* for 9 min and resuspended in Lonza electroporation P3 buffer with supplement (20 µl per 1 × 10⁶ T cells). The T cells (20 µl) were electroporated with 3.5 µl ribonucleoprotein and 1 µg TCR-encoding plasmid DNA (1–2 µl) using a Lonza 4D Nucleofector 96-well electroporation system and pulse code EH115 (ref. [61]). CD8[+] T cells were immediately rescued by the addition of 80 µl warmed T cell medium and incubation in a 37 °C incubator for 15 min. The cells were then split into fifths in 96-well round-bottomed plates; T cell medium plus 10 ng ml⁻¹ IL-2 was added to the samples to a final volume of 200 µl. The CD8[+] T cells were expanded for a minimum of 96 h before testing for pMHC tetramer binding. The T cells were re-fed with a half volume of fresh medium and IL-2 every 3–4 days.

## pMHC tetramer screening

Ultraviolet photolabile pMHC I monomers for HLA-A*01:01, HLA-A*2:01, HLA-A*03:01, HLA-A*24:02, HLA-B*08:01, HLA-B*15:01, HLA-B*35:01 and HLA-B*44:02 were obtained from the NIH Tetramer Core. Custom peptide-loaded MHC I monomers were generated by ultraviolet light–ligand exchange as previously described[62]. HLA-A*31:01 pMHC monomers (Easymers) were purchased from ImmunAware and loaded with custom peptides according to the supplier's instructions. Tetramerization was carried out using streptavidin conjugated to the fluorophores phycoerythrin and allophycocyanin (Life Technologies). CD8[+] T cells were treated with 100 nM dasatinib (StemCell) for 30 min at 37 °C, followed by staining with the appropriate tetramers (2–3 µg ml⁻¹) for 30 min at room temperature. All tetramers were used within 3–4 weeks of synthesis. The cells were washed in FACS buffer (1×DPBS without calcium or magnesium, 0.1% sodium azide, 2 mM EDTA and 1% FBS) and stained with anti-CD8 PECy7 (eBioscience; SK1), anti-TCR BV421 (BioLegend; IP26), a PerCP/Cy5.5 dump antibody mixture containing anti-CD4 (BioLegend; RPA-T4), anti-CD14 (BioLegend; HCD14), anti-CD16 (BioLegend; B73.1), anti-CD19 (BioLegend; HIB19; all antibodies at 1:100) and Aqua506 viability dye (1:1,000; Life Technologies) for 30 min at 4 °C. The cells were then washed and resuspended in FACS buffer, and analyzed by flow cytometry (LSRFortessa). Only experiments where the forward versus side-scatter gate contained at least 10% lymphocytes and CD8[+] T cells expressed fewer than 20% TCRs were used for analysis to ensure a large number of T cells with high TCR-knockout efficiency was achieved (Extended Data Fig. 5a).

## pMHC yeast display selection

Yeast libraries were developed as previously described[19]. Yeast allele libraries were thawed in SDCAA (pH 5) medium, passaged, induced in SGCAA (pH 5) and selected using biotinylated soluble TCR coupled to streptavidin-coated magnetic MACS beads (Miltenyi) as previously described[63]. Briefly, 2 × 10⁹ yeast cells from all four length libraries underwent negative selection with 250 µl beads in 5 ml PBE (PBS containing 0.5% BSA and 1 mM EDTA) for 1 h with rotation at 4 °C. After passage through an LS column (Miltenyi) on a magnetic stand and three washes with 3 ml PBE, the flow-through was incubated with 250 µl beads (pre-incubated with 400 nM biotinylated TCR) for 3 h at 4 °C with rotation. The yeast were magnetically separated through an additional LS column, washed three times with 3 ml PBE and the elution was cultured overnight in SDCAA (pH 5) following an SDCAA wash to remove residual PBE. The yeast were induced in SGCAA (pH 5) for 2–3 days before further selection, with subsequent selections using 50 µl beads or TCR-coated beads in 500 µl PBE.

## Deep sequencing of pMHC yeast libraries

DNA was isolated from $5–10 \times 10^7$ yeast cells per selection using a Zymoprep II kit (Zymo Research). Unique barcodes and random eight-mer sequences were added to the sequencing product by PCR and amplified for 25 cycles to allow for downstream demultiplexing and improved clustering. A subsequent PCR added Illumina chip primer sequences, resulting in products containing Illumina P5-Truseq read 1-(N8)-Barcode-pHLA-(N8)-Truseq read 2-IlluminaP7. The library was purified by double-sided SPRI bead isolation (Beckman Coulter), quantified using a KAPA library amplification kit (Illumina) and deep sequenced on an Illumina MiSeq instrument with a $2 \times 150$ V2 kit for low-diversity libraries.

## Generation of HLA-expressing APC lines

The genes for HLA-A*31:01, HLA-A*29:02, HLA-B*08:01 and HLA-B*35:01 were codon-optimized and synthesized as gene blocks (IDT). The gene blocks were cloned into the pHR-CMV Lacz lentivirus vector by Gibson assembly and sequences were verified by Sanger sequencing. The vector was expressed in K562 cells by lentiviral transduction and selected by puromycin.

## Cytokine assays

The APCs for all cytokine assays were T2 cells expressing HLA-A*02:01, HLA-A*01:01 or HLA-A*03:01, or K562 cells expressing HLA*29:02, HLA-A*31:01, HLA-B*08:01 or HLA-B*35:01. The APCs were pulsed overnight with 10 µg ml$^{-1}$ peptide or vehicle control in serum-free medium. CD8$^+$ T cells ($2 \times 10^5$) were stimulated with peptide-loaded APCs ($1 \times 10^5$ per condition) for 6 h in the presence of 1:500 GolgiStop (BD), 1:500 GolgiPlug (BD) and 1:200 CD28/CD49d (FastImmune; BD). The cells were washed with FACS buffer and stained with the cell surface antibodies as described in the 'pMHC tetramer screening' section (anti-CD8, anti-TCR, dump channel antibody mixture and live/dead dye). Next, the cells were washed, fixed and stained with anti-human IFN-γ Alexa 647 (BioLegend; 4S.B3) and anti-human TNF-α Alexa 488 (BioLegend; Mab11) in permeabilization buffer (BD). Finally, the cells were washed and collected on an LSRFortessa system.

## Generation of TCR-expressing NFAT−mCherry Jurkat cells

Jurkat E6-1 T cells (American Type Culture Collection, TIB-152) were maintained in RPMI medium supplemented with L-glutamine and 10% FBS. Endogenous *TRAC* and *TRBC1* expression in Jurkat cells were knocked out with synthetic crRNAs designed using the Alt-R system (IDT) containing the following genomic target sequences: *TRBC1*, 5′-CGTAGAACTGGACTTGACAG-3′ and *TRAC*, 5′-CTTCAAGAGCAACAGTGCTG-3′. The crRNA was complexed with 1:1 *trans*-activating crRNA (IDT; 0.2 nmol each), followed by 0.1 nmol recombinant Cas9 protein (Macrolab). The ribonucleoproteins were then transduced into Jurkat T cells using a Amaxa P3 primary cell nucleofector kit (Lonza; pulse code CK116). *TRAC* knockout was performed first and loss of surface TCRαβ expression was confirmed by flow cytometry. *TRAC*-knockout cells underwent subsequent knockout of *TRBC1*, which had previously been shown to lead to loss of TCRαβ expression in line with overexpressed TCRα. To track TCR activation, a lentiviral vector was constructed that contained the NFAT transcriptional reporter NBV[64] upstream of a minimal CMV reporter driving mCherry fluorescent marker expression, and constitutive expression of iRFP670 under a Pgk promoter provided a marker of transduction. Jurkat cells lacking endogenous TCRαβ expression were transduced with the vector and sorted for iRFP fluorescence and lack of mCherry background fluorescence. For TCRs corresponding to CD8$^+$ T cells, an additional lentiviral vector encoding human CD8α was expressed in the Jurkat cells and the cells were sorted for uniform CD8α expression before TCR transduction. For TCR expression, lentiviral expression constructs that encode a human Pgk promoter and the coding sequence of each specific TCRα chain with the IRES-neomycin resistance gene or each

TCRβ chain with the IRES-blasticidin resistance gene were generated. Lentiviral particles were packaged in HEK293T cells following standard protocols and concentrated 10× using the Lenti-X Concentrator reagent (Takara). Viral particles were added to T cells at a low multiplicity of infection and expression was ensured by passaging cells for 5 days under antibiotic selection with 10 µg ml$^{-1}$ blasticidin (Gibco) and 1 mg ml$^{-1}$ G418 (Teknova).

## TCR-expressing Jurkat cell assays

TCR-expressing Jurkat cells ($1 \times 10^5$) were stimulated for 24 h with HLA-allele-transduced APCs ($1 \times 10^5$) loaded with 10 µg ml$^{-1}$ peptide or vehicle control. Antigen-reactive CD8$^+$ cells were identified by coexpression of NFAT−mCherry and anti-human CD69−phycoerythrin (BioLegend; FN50).

## Generation of lymphoblastoid cell lines

A Pan B cell isolation kit (Miltenyi) was used to isolate B cells from frozen PBMCs. The B cells ($1 \times 10^6$ cells ml$^{-1}$) were incubated 1:1 with pre-warmed EBV supernatant (B95.8 strain) for 1 h at 37 °C, followed by the addition of 1 µg ml$^{-1}$ R848 and 100 ng ml$^{-1}$ CD40L. The cells were cultured for two weeks, with medium changes as needed, and expanded into larger plates. The LCLs were cryopreserved for future use in cellular assays.

## TCR-expressing Jurkat cell co-culture with lymphoblastoid cell lines

Jurkat cells expressing the TCR of interest (target TCR Jurkat cells) were co-cultured with LCLs carrying at least one matching MHC I allele (HLA-matched LCLs) at a 1:1 ratio (100,000 cells each per well) in a 96-well plate for 24 h at 37 °C. To assess specificity, negative control conditions included co-culture of (1) target TCR Jurkat cells with HLA-mismatched LCLs and (2) HLA-matched LCLs with Jurkat cells expressing an HLA-mismatched TCR. Jurkat reactivity was assessed by measurement of coexpression of NFAT−mCherry and CD69−phycoerythrin as described in the 'TCR-expressing Jurkat cell assays' section.

## DNA and RNA extraction from cerebrospinal fluid

Cell-free CSF supernatant was obtained after centrifugation as described earlier for single-cell sequencing and stored at −80 °C. DNA and RNA were each extracted from 400 µl CSF supernatant using a ZYMO Quick-DNA/RNA pathogen MagBead kit (Zymo Research) according to the manufacturer's instructions. The extracted DNA was eluted in 50 µl nuclease-free water. DNase I treatment was performed on the RNA-extraction samples before elution into 30 µl nuclease-free water. An EBV-transformed LCL was used as a positive control; Jurkat cells and water only were used as negative controls. The DNA concentration and purity were assessed using a Nanodrop spectrophotometer (Thermo Fisher Scientific). The extracted DNA and RNA were stored at −20 °C until use for PCR amplification. Complementary DNA was synthesized using a ProtoScript first strand cDNA synthesis kit (NEB) using 6 µl RNA per 20 µl reaction volume according to the manufacturer's instructions.

## PCR amplification of cerebrospinal fluid DNA

The following previously described[65] primers (IDT) for the *BZLF1* promoter were used: 5′-AGCATGCCATGCATATTTC-3′ (forward) and 5-TTGGCAAGGTGCAATGTTT-3′ (reverse). PCR reactions were performed using a Qiagen Taq PCR core kit according to the manufacturer's instructions. Each 30-µl reaction contained 1×PCR Buffer with MgCl$_2$, 200 µM dNTPs, 0.2 µM forward and reverse primers, 10–100 ng template DNA, 0.75 U Qiagen Taq DNA polymerase and nuclease-free water to the final volume. Amplification was carried out in a thermal cycler (Bio-Rad thermal cycler with 96-deep-well C1000 block) using the following cycling conditions: an initial denaturation step at 95 °C for 3 min, followed by 34 cycles of denaturation at 95 °C for 30 s, annealing at 58 °C for 30 s and extension at 72 °C for 1 min. A final extension

step at 72 °C for 5 min was performed, followed by a hold at 4 °C. The PCR products were analyzed using agarose gel electrophoresis and visualized under ultraviolet light using a gel documentation system (LI-COR Odyssey M Imager). For samples showing positive amplification, the remaining PCR reaction volume was directly submitted to Molecular Cloning Laboratories for Sanger sequencing. CSF samples were considered EBV-positive if sequencing results correctly aligned to the reference sequence of the amplified target.

## Analysis of cerebrospinal fluid DNA and cDNA using ddPCR

Primers and probes for ddPCR were synthesized as PrimeTime qPCR assays (IDT). Probes targeting EBV genes were labeled with FAM and probes for the housekeeping reference genes (*RPP30* or *GAPDH*) were labeled with HEX (Supplementary Table 22). Oligonucleotides were used as previously published for *EBER2* (ref. [66]), *BamHI-W*[67] and *RPP30* (ref. [68]). A volume of 1.25 µl of each 20× target primers–probe mix in Tris–EDTA was used with 2×ddPCR Supermix for probes without deoxyuridine triphosphate and 10 µl DNA. A total reaction volume of 20 µl was loaded with a 70-µl oil droplet using a QX100 droplet generator (Bio-Rad). The emulsion of approximately 40 µl was slowly transferred to ddPCR 96-well plates (Bio-Rad, 12001925) and heat-sealed with foil. Amplification was carried out in a thermal cycler (Bio-Rad Thermal Cycler with 96-Deep Well C1000 block) using the following cycling conditions: an initial denaturation step at 95 °C for 10 min; 39 cycles of 30 s at 94 °C and 1 min at 57 °C, followed by 10 min at 98 °C. Analysis of the ddPCR data was performed using the Bio-Rad QuantaSoftTM software. Data from the droplet reader are given as copies per microlitre and relative expression was calculated as the target gene:reference gene ratio. Any sample that was not detected by the housekeeping gene was repeated and the threshold was set separately according to the negative control with water. All samples were run in duplicate.

## Statistical analyses

Differential gene expression comparisons between groups were performed using the two-sided Wilcoxon rank-sum test with Bonferroni correction. Shannon entropy results were compared using Brown–Forsythe's and Welch's analysis of variance with multiple comparisons using Dunnett T3 corrections. Comparisons of CSF-enriched clonotypes T cell (for example, CD8+ versus CD4+ T cells and MS/CIS versus HC/OND) and ddPCR results were performed using unpaired two-tailed Student's *t*-tests with Welch's correction using GraphPad Prism (v10.6.1).

## Reporting summary

Further information on research design is available in the Nature Portfolio Reporting Summary linked to this article.

## Data availability

All data are available in the main text or the supplementary materials. The scRNA-seq data have been uploaded to the Gene Expression Omnibus (GEO) repository under BioProject PRJNA549712 (GEO accession number GSE133028) and the scTCR-seq data have been uploaded under BioProject PRJNA1232831 at GEO accession number GSE291328. Source data are provided with this paper.

## Code availability

The code for all scRNA-seq and scTCR-seq analysis can be found at https://github.com/UCSF-Wilson-Lab/MS_Tcell_CSF_PBMC_single_cell_study_analysis.

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

## Acknowledgements

We thank the individuals who agreed to participate as research subjects in this study. We thank the UCSF EPIC and Origins Study Teams for valuable aid in subject recruitment. We acknowledge funding from the Japan Society for the Promotion of Science (grant number 202370027; F.H.), Japanese Society of Neurology (F.H.), Mayer Foundation (J.J.S.), National Institutes of Health (grant numbers K08NS107619 to J.J.S., R01AI158861 and R01AI169070 to J.A.H., R01NS092835 and R35NS111644 to S.L.H. and M.R.W., and R21AI142186 to M.R.W. and S.S.Z.), National Multiple Sclerosis Society (grant numbers FAN-1608-25607 to R.D.S., and FAN-1506-04555 and RG-2110-38434 to J.J.S.), Race to Erase MS (J.J.S.), Uehara Memorial Foundation (grant number 202140013; F.H.), Valhalla Foundation (S.L.H. and B.A.C.C.) and Westridge Foundation (M.R.W.).

## Author contributions

Conceptualization: B.A.C.C., S.L.H., S.S.Z., M.R.W. and J.J.S. Participant recruitment: K. Koshal, T.C., M.H. and R.G. Methodology: F.H., K.M., R.D., J.G., E.H., R.D.S., G.H., A.R., I.J.F., C.D.C., J.A.H., M.G. and J.J.S. Investigation: F.H., K.M., R.D., J.G., E.H., R.D.S., L.O., R.L., A.G., D.G.A., G.H., S.A., A.S., A.R., E.T., K. Koshal, K. Kizer, J.D., A.K.C., F.S., L.F., T.M., L.L.K., I.J.F., C.D.C. and L.S. Visualization: R.D. and J.J.S. Funding acquisition: J.J.S., B.A.C.C., S.L.H., M.R.W. and S.S.Z. Supervision: J.J.S., S.S.Z. and M.R.W. Writing—original draft and revision: J.J.S. Writing—review and editing: B.A.C.C., S.S.Z. and M.R.W.

## Competing interests

A.R. is a current employee of Genentech. J.D., A.K.C., F.S., L.F., T.F., L.L.K., L.S. and M.G. are currently or previously employed by 3T Biosciences. M.R.W. has received research grant funding from Roche/Genentech, Novartis and Kyverna Therapeutics; speaking honoraria from Genentech, Takeda, WebMD and Novartis; and consulting fees from Vertex Pharmaceuticals, Ouro Medicines, Indapta Therapeutics, Pfizer and Delve Bio; is on the Board of Directors of Delve Bio and has received licensing fees from CDI Labs. J.J.S. has received research grant funding from Roche/Genentech and Novartis, advisory board honoraria from IgM Biosciences and TG Therapeutics, and has received stock options as a consultant for Sift BioSciences. The remaining authors declare no competing interests.

## Additional information

**Extended data** is available for this paper at https://doi.org/10.1038/s41590-025-02412-3.

**Correspondence and requests for materials** should be addressed to Scott S. Zamvil or Joseph J. Sabatino.

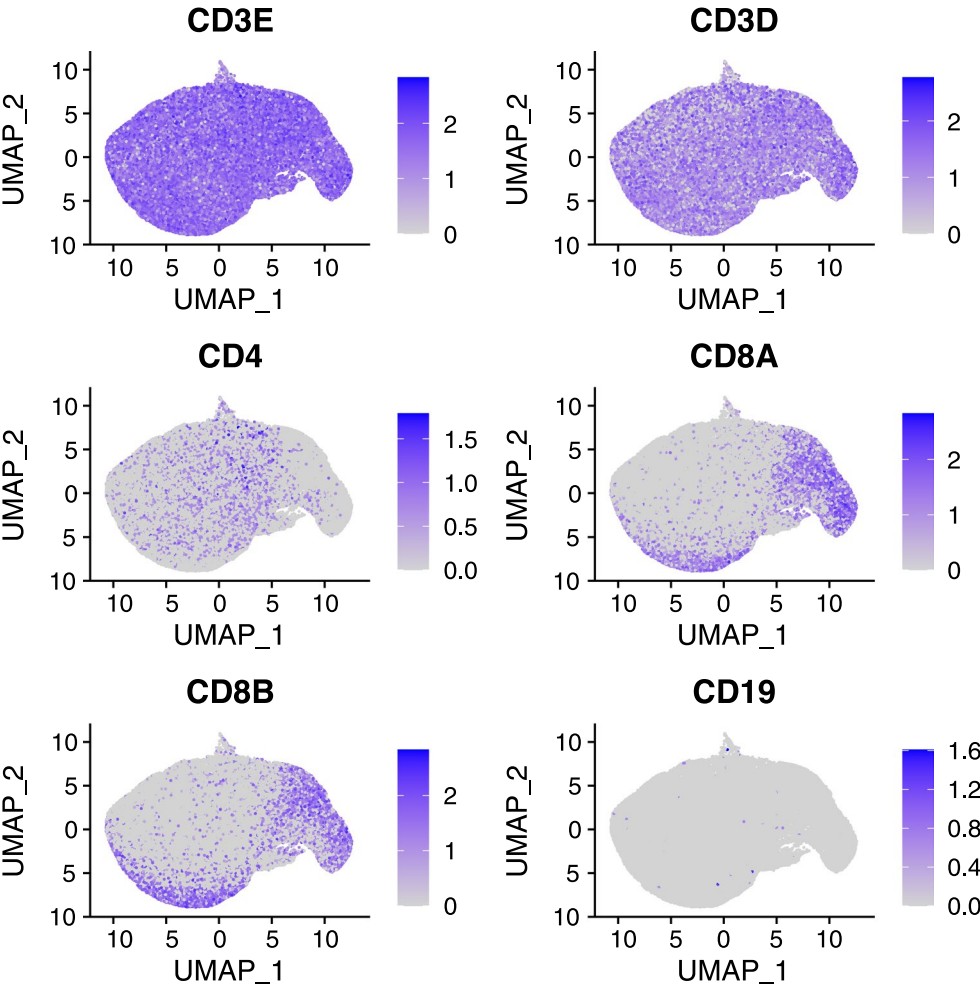

**Extended Data Fig. 1 | T cell gene expression analysis.** Expression for the indicated genes is shown for all T cells (blood and CSF combined) after merging scRNA-seq and scTCR-seq data.

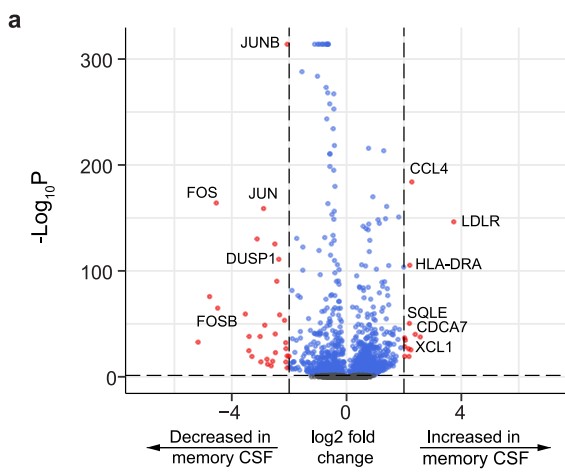

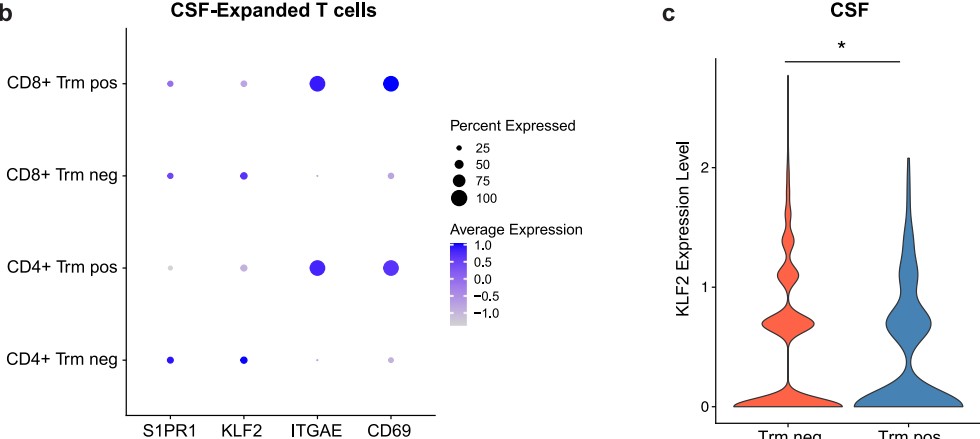

**Extended Data Fig. 2 | Memory T cell gene expression. a**, Volcano plot analysis of differential gene expression between memory (CD27⁺) T cells in the CSF and blood. Only genes expressed in at least 10% of memory CD8⁺ T cells in the blood and/or CSF were analyzed. Differential gene expression comparisons were performed using two-sided Wilcon ranked-sum tests with Bonferroni correction for adjusted p-values. Genes with adjusted p-values < 0.05 and log₂ fold change > 2 are indicated in red and genes with adjusted p-values < 0.05 in blue. **b**, Expression levels and percent expression of the indicated genes is shown for tissue-resident memory (Trm) positive (CD69⁺ and ITGAE⁺) and Trm negative CD4⁺ and CD8⁺ T cells. **c**, KLF2 expression levels between Trm negative and positive T cells in the CSF. *p < 0.05.

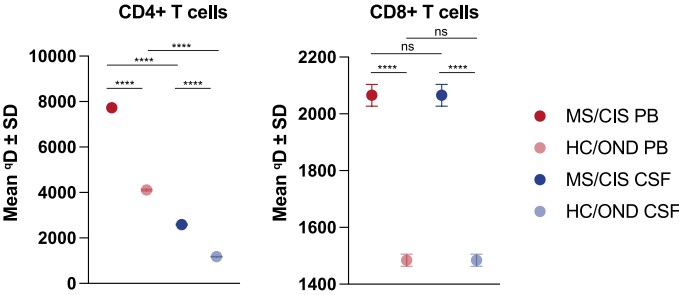

**Extended Data Fig. 3 | T cell diversity analysis.** Shannon entropy analysis (where *y* axis indicates exponential of Shannon–Wiener index) is shown for CD8[+] T cells in the peripheral blood (PB) and CSF by disease status. Abbreviations: MS = multiple sclerosis; CIS = clinically isolated syndrome; HC = healthy control; OND = other neuro-inflammatory disease. Comparisons between groups using Brown–Forsythe and Welch ANOVA with multiple comparisons using Dunnett T3 corrections (****p < 0.0001; ns = not significant).

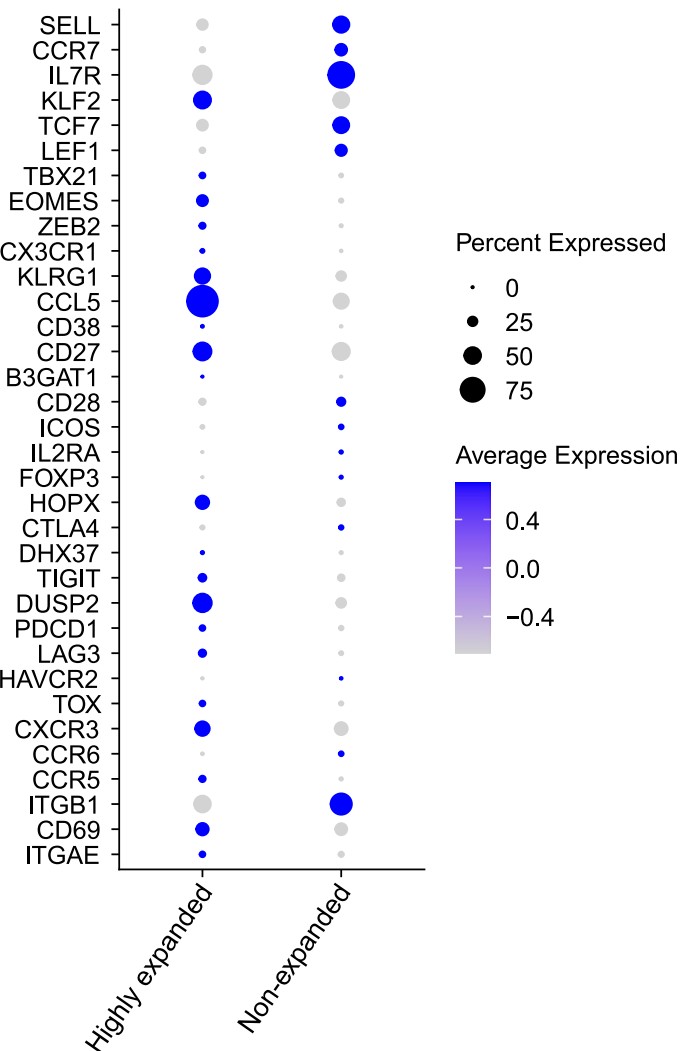

**Extended Data Fig. 4 | Gene expression analysis of CSF T cells by expansion status.** Expression levels and percent expression of the indicated genes is shown for highly expanded CSF T cells versus non-expanded T cell clonotypes.

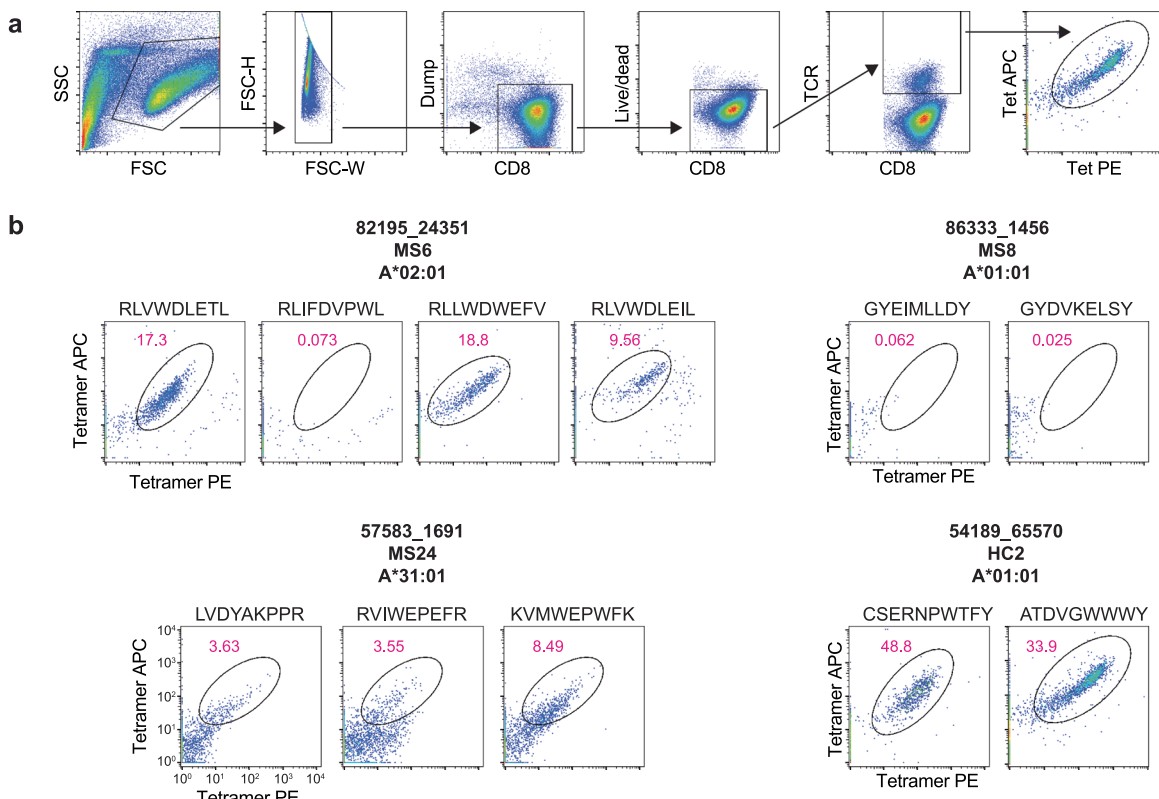

**Extended Data Fig. 5 | Tetramer screening of antigens identified by pMHC yeast display. a**, Representative flow cytometry analysis of pMHC tetramer-stained CD8+ T cells following TCR knock-in. **b**, Four patient-derived TCRs were tested for tetramer binding to the indicated peptides.

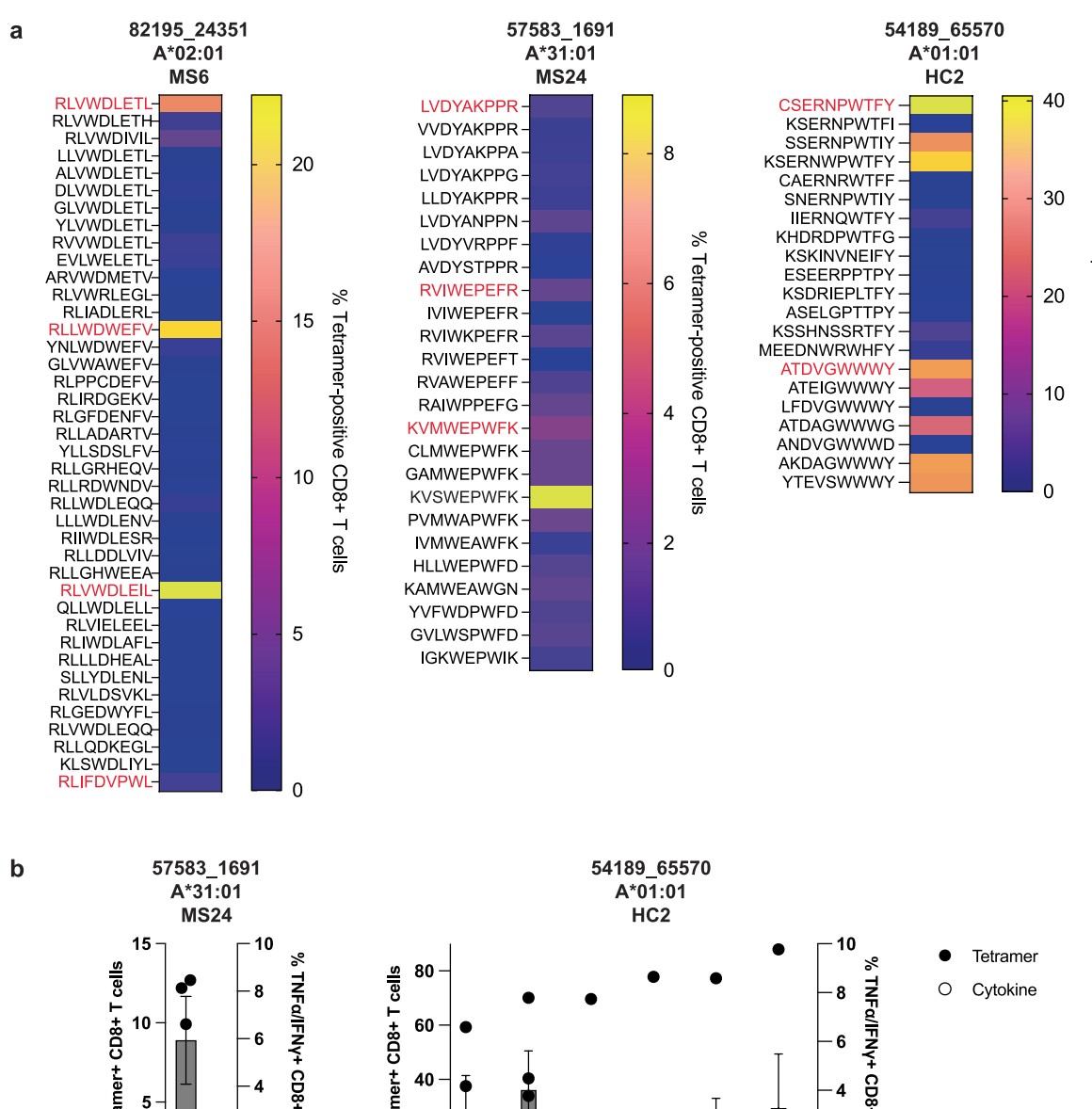

**Extended Data Fig. 6 | Validation results of peptide homologs to yeast display mimotopes. a**, Summary tetramer binding analysis of peptide homologs for the three TCRs that yielded yeast display-derived mimotopes (red). **b**, Summary of tetramer binding and cytokine reactivity of the indicated peptides to two different TCRs. Cytokine reactivity reflects subtracted background from no-stimulation control. Each peptide was tested a minimum of two times using T cells from different donors for tetramer and cytokine, respectively (KVSWEPWFK $n = 4$ and $n = 2$, SSERNPWTIY $n = 4$ and $n = 2$, KSERNWPWTFY $n = 4$ and $n = 2$, ATEIGWWWY $n = 6$ and $n = 2$, ATDAGWWWG $n = 6$ and $n = 2$, AKDAGWWWY $n = 6$ and $n = 4$, YTEVSWWWY $n = 4$ and $n = 4$).

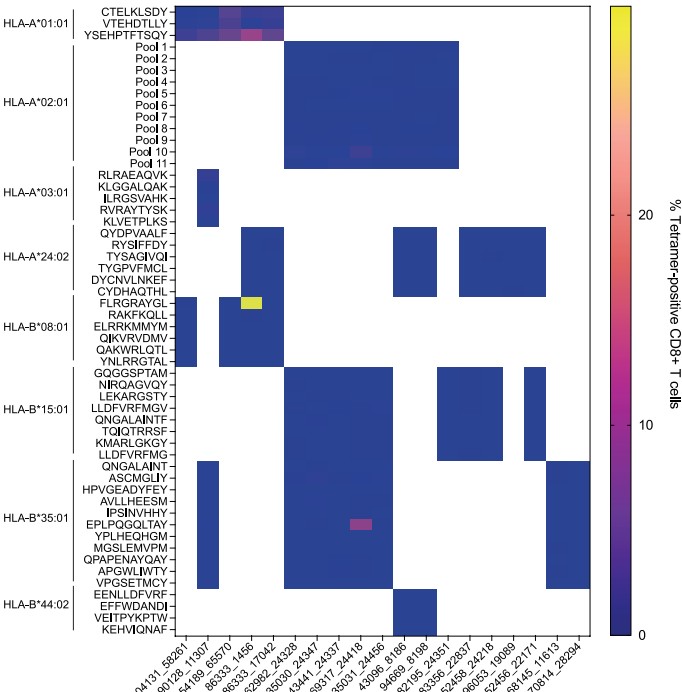

**Extended Data Fig. 7 | Summary of viral antigen specificity of highly expanded CSF-enriched CD8⁺ T cells.** The summary of all pMHC tetramer screening for 98 viral peptides for 19 patient-derived TCRs is shown in the heatmap. TCR clonotype ID's are shown on the x axis and peptide:MHC antigens are shown on the y axis. All viral peptides were tested individually except in the case of the peptides for HLA-A*02:01 where tetramers were tested in pools of 5. Each peptide/pool was tested a minimum of two times in using different T cells from different donors.

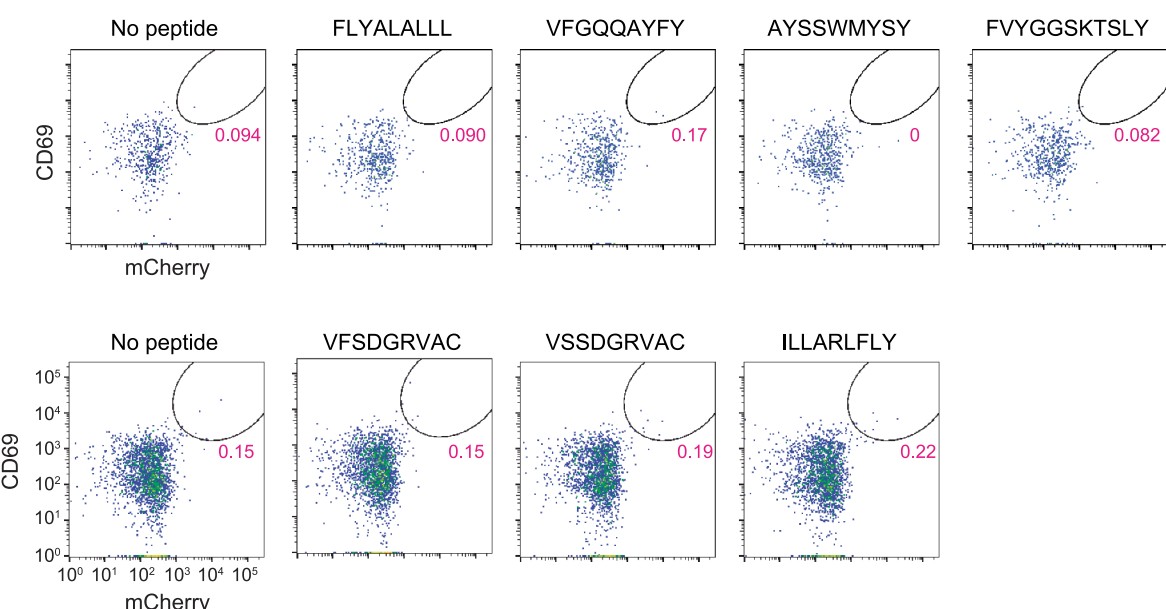

**Extended Data Fig. 8 | HLA-A*29:02 EBV peptide testing.** Representative flow cytometry analysis of Jurkat reporter cells expressing TCR 94669_8198 were co-cultured with HLA-A*29:02-expressing K562 cells pulsed with the indicated EBV peptides or no peptide for 24 h (all tested in triplicate). Antigen reactivity was assessed by coexpression of CD69 and NFAT–mCherry.

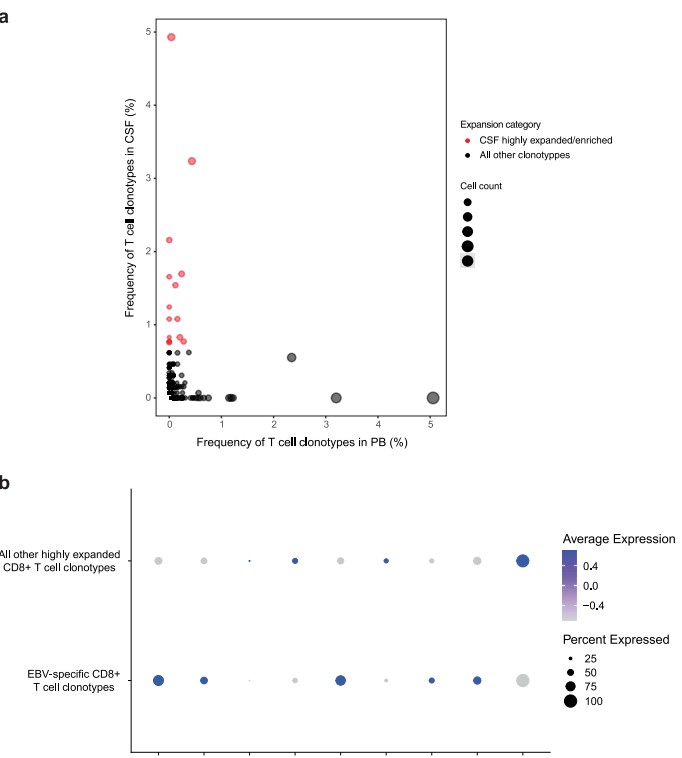

**Extended Data Fig. 9 | EBV-specific CD8⁺ T cell clonal analysis. a**, The TCR clonotype frequencies for all clonotypes of the three MS patients (MS6, MS8, MS27) with CSF-expanded EBV-specific CD8⁺ T cells was compared between the blood and CSF. Clonal frequency of T cell clonotypes in the CSF that were highly expanded (at least 0.75%) and enriched at least 2-fold more frequently than the blood of the same individual are highlighted in red. **b**, The expression of specific genes between the three CSF-expanded EBV-specific CD8⁺ T cell clonotypes (TCRs 86333_1456, 69317_24418, and 94669_8198 combined) and all other CSF-expanded enriched CD8⁺ T cells was compared.

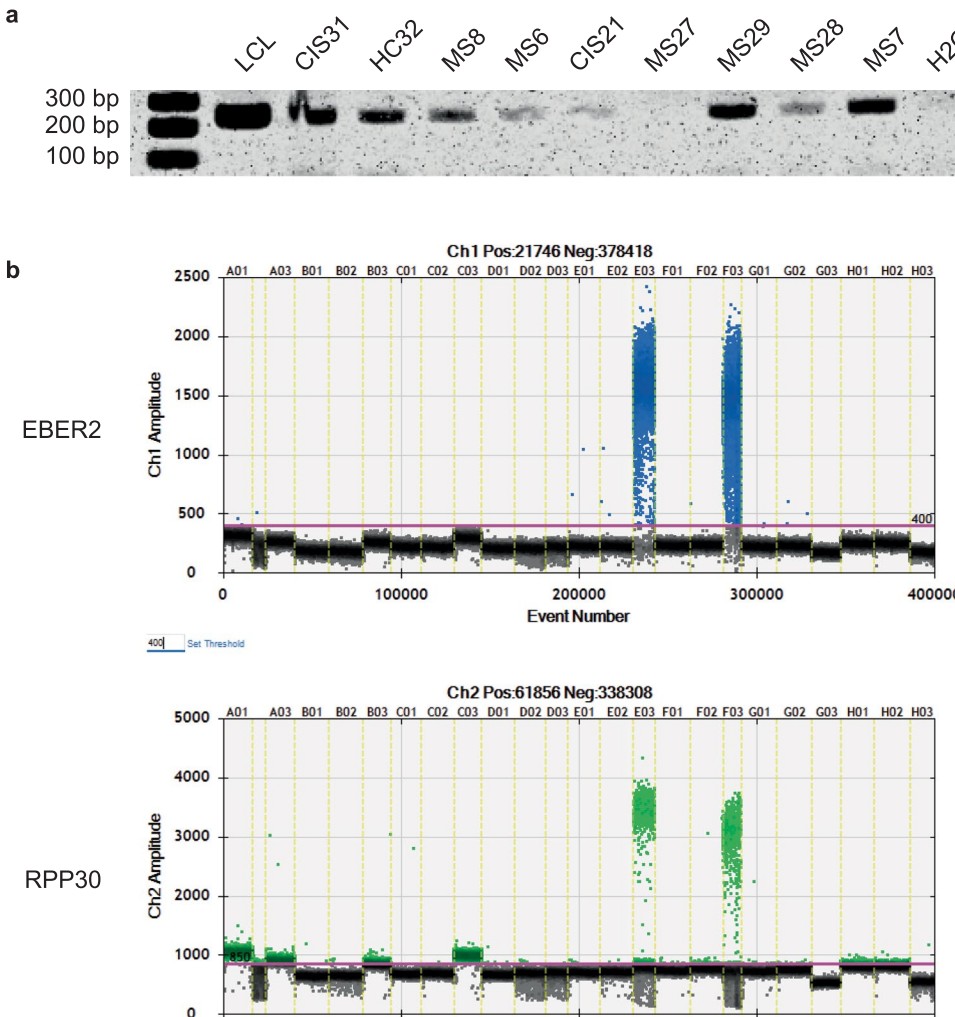

**Extended Data Fig. 10 | EBV DNA PCR amplification in CSF. a**, Representative agarose gel of *BZLF1* DNA amplification (239-bp fragment) in the CSF supernatant of the indicated patient samples. DNA from EBV-transformed lymphoblastoid cell lines (LCL) were used as positive control and water as a negative control. Samples with a positive band underwent Sanger sequencing for confirmation. **b**, Representative DNA ddPCR results of *EBER2* and *RPP30* (housekeeping gene) from CSF supernatant is shown. The two columns with many positive droplets with high signal are from LCLs as positive control. The positive threshold (purple line) was defined as the signal above the water negative control.

# Reporting Summary

## Statistics

For all statistical analyses, confirm that the following items are present in the figure legend, table legend, main text, or Methods section.

| n/a | Confirmed | |
|---|---|---|
| ☐ | ☒ | The exact sample size (*n*) for each experimental group/condition, given as a discrete number and unit of measurement |
| ☒ | ☐ | A statement on whether measurements were taken from distinct samples or whether the same sample was measured repeatedly |
| ☐ | ☒ | The statistical test(s) used AND whether they are one- or two-sided *Only common tests should be described solely by name; describe more complex techniques in the Methods section.* |
| ☒ | ☐ | A description of all covariates tested |
| ☐ | ☒ | A description of any assumptions or corrections, such as tests of normality and adjustment for multiple comparisons |
| ☐ | ☒ | A full description of the statistical parameters including central tendency (e.g. means) or other basic estimates (e.g. regression coefficient) AND variation (e.g. standard deviation) or associated estimates of uncertainty (e.g. confidence intervals) |
| ☐ | ☒ | For null hypothesis testing, the test statistic (e.g. *F*, *t*, *r*) with confidence intervals, effect sizes, degrees of freedom and *P* value noted *Give P values as exact values whenever suitable.* |
| ☒ | ☐ | For Bayesian analysis, information on the choice of priors and Markov chain Monte Carlo settings |
| ☒ | ☐ | For hierarchical and complex designs, identification of the appropriate level for tests and full reporting of outcomes |
| ☒ | ☐ | Estimates of effect sizes (e.g. Cohen's *d*, Pearson's *r*), indicating how they were calculated |

*Our web collection on statistics for biologists contains articles on many of the points above.*

## Software and code

Policy information about availability of computer code

| | |
|---|---|
| Data collection | The software used for data collection is described in the relevant portions of the Materials and Methods. |
| Data analysis | The software and their versions used for data analysis listed below and are described in the relevant portions of the Materials and Methods. CellRanger v3.0.1 (scRNA-seq) and v3.1.0 (scTCR-seq) Seurat v3.1.2-v4.3.0 Spliced Transcripts Alignment to a Reference (STAR) algorithm v2.5.1 SingleR v1.1.7 DoubletFinder v2.0.2 Immcantation v3.1.0 Change-O v0.4.6 GraphPad Prism v10.6.1  All custom code used for the analysis of data and generation of plots is available upon request. |

For manuscripts utilizing custom algorithms or software that are central to the research but not yet described in published literature, software must be made available to editors and reviewers. We strongly encourage code deposition in a community repository (e.g. GitHub). See the Nature Portfolio guidelines for submitting code & software for further information.

## Data

Policy information about availability of data

All manuscripts must include a data availability statement. This statement should provide the following information, where applicable:

- Accession codes, unique identifiers, or web links for publicly available datasets
- A description of any restrictions on data availability
- For clinical datasets or third party data, please ensure that the statement adheres to our policy

All scRNA-seq data is available on BioProject PRJNA549712 (GEO accession no. GSE133028). All scTCR-seq data will be made publicly available without restrictions on BioProject PRJNA1232831 effective January 1, 2026 (GEO accession no. GSE291328).

## Research involving human participants, their data, or biological material

Policy information about studies with human participants or human data. See also policy information about sex, gender (identity/presentation), and sexual orientation and race, ethnicity and racism.

| | |
|---|---|
| Reporting on sex and gender | Sex data refers to biologic attributes of research participants. |
| Reporting on race, ethnicity, or other socially relevant groupings | N/A |
| Population characteristics | Age, sex, diagnosis, MRI and CSF information, and treatment status were provided in Table 1 and Supplemental Table 1 of the Manuscript. |
| Recruitment | MS/CIS and control participants were enrolled through the University of California San Francisco (UCSF) ORIGINS or Expression, Proteomics, Imaging, Clinical (EPIC) studies (https://epicstudy.ucsf.edu/). This study is designed to enroll patients early after experiencing an acute CNS demyelinating event. Healthy controls and OND patients were enrolled in the biobanking study "Immunological Studies of Neurologic Subjects". Informed consent was obtained from all participants in this study. No compensation was provided to study participants. We are not aware of any self-selection bias which would alter the results of this study as all patients and controls meeting eligibility criteria were able to enroll. |
| Ethics oversight | The studies in this manuscript have been approved by the UCSF IRB research ethics committee (protocol numbers 10-02389 and 14-15278). |

Note that full information on the approval of the study protocol must also be provided in the manuscript.

# Field-specific reporting

Please select the one below that is the best fit for your research. If you are not sure, read the appropriate sections before making your selection.

☒ Life sciences ☐ Behavioural & social sciences ☐ Ecological, evolutionary & environmental sciences

For a reference copy of the document with all sections, see nature.com/documents/nr-reporting-summary-flat.pdf

# Life sciences study design

All studies must disclose on these points even when the disclosure is negative.

| | |
|---|---|
| Sample size | Sample size was determined by sample availability. No sample size calculations were performed. Given the disproportionate number of participants in different disease categories, we grouped MS and CIS patients together (n = 13) and against OND and HC together as a comparison non-MS group (n = 5). |
| Data exclusions | No data were excluded. Specific criteria were employed for quality control and analysis of scRNA-seq and scTCR-seq data as described in Materials and Methods. |
| Replication | All flow cytometry experiments were performed in a minimum of two independent experiments. ddPCR experiment results were performed in duplicate. |
| Randomization | Samples were allocated based on clinical disease category. |
| Blinding | Researchers were not blinded during sample acquisition and data analysis as it was important for the investigators to know the disease status of the subjects in order to complete the data analysis. |

# Reporting for specific materials, systems and methods

We require information from authors about some types of materials, experimental systems and methods used in many studies. Here, indicate whether each material, system or method listed is relevant to your study. If you are not sure if a list item applies to your research, read the appropriate section before selecting a response.

## Materials & experimental systems

| n/a | Involved in the study |
|-----|------------------------|
| ☐ | ☒ Antibodies |
| ☐ | ☒ Eukaryotic cell lines |
| ☒ | ☐ Palaeontology and archaeology |
| ☒ | ☐ Animals and other organisms |
| ☒ | ☐ Clinical data |
| ☒ | ☐ Dual use research of concern |
| ☒ | ☐ Plants |

## Methods

| n/a | Involved in the study |
|-----|------------------------|
| ☒ | ☐ ChIP-seq |
| ☐ | ☒ Flow cytometry |
| ☒ | ☐ MRI-based neuroimaging |

## Antibodies

| | |
|---|---|
| Antibodies used | Anti-human CD8 PECy7 (eBioscience; SK1), anti-human TCR BV421 (BioLegend; IP26), anti-human CD4 PerCP-Cy5.5 (BioLegend; RPA-T4), anti-CD14 PerCP-Cy5.5 (BioLegend; HCD14), anti-human CD16 PerCP-Cy5.5 (BioLegend; B73.1), anti-human CD19 PerCP-Cy5.5 (BioLegend; HIB19), anti-human IFNg Alexa 647 (BioLegend; 4S.B3),  anti-human TNFα Alexa 488 (BioLegend; Mab11), and anti-human CD69 PE (BioLegend; FN50).  All antibodies were used at 1:100 dilutions. |
| Validation | All antibodies were purchased from the indicated commercial vendors with validation data and applicable citations available on product listings. |

## Eukaryotic cell lines

Policy information about cell lines and Sex and Gender in Research

| | |
|---|---|
| Cell line source(s) | Jurkat E6-1 T cells (ATCC TIB-152)<br>293T cell line (ATCC CRL-3216)<br>K562 cell line (ATCC CCL-243)<br>Lymphoblastoid cell lines (LCLs) - EBV-transformed primary B cells from MS patients or healthy controls. |
| Authentication | None of the cell lines were authenticated. |
| Mycoplasma contamination | The cell lines were not tested for mycoplasma contamination. |
| Commonly misidentified lines<br>(See ICLAC register) | K562 cells |

## Plants

| | |
|---|---|
| Seed stocks | N/A |
| Novel plant genotypes | N/A |
| Authentication | N/A |

## Flow Cytometry

### Plots

Confirm that:

☒ The axis labels state the marker and fluorochrome used (e.g. CD4-FITC).

☒ The axis scales are clearly visible. Include numbers along axes only for bottom left plot of group (a 'group' is an analysis of identical markers).

☒ All plots are contour plots with outliers or pseudocolor plots.

☒ A numerical value for number of cells or percentage (with statistics) is provided.

# Methodology

| | |
|---|---|
| Sample preparation | For pMHC I tetramer analysis: Primary human CD8+ T cells expressing TCRs of interest were generated by CRISPR knockin as described in Materials and Methods.  CD8+ T cells were treated wisdfsdfdth 100 nM dasatinib (StemCell) for 30 min at 37 °C followed by staining with the appropriate tetramers (2-3 µg/mL) for 30 min at room temperature. Cells were washed in FACS buffer and stained with the indicated cell surface antibodies for 30 minutes at 4°C.  Cells were then washed and resuspended in FACS buffer and analyzed by flow cytometry.<br><br>For intracellular cytokine staining: APCs for were pulsed with 10 µg/ml peptide or vehicle control overnight in serum-free media.  CD8+ T cells (2 x 105) were stimulated with peptide-loaded APCs (1 x 105 per condition) for 6 hours in the presence of 1:500 GolgiStop (BD), 1:500 GolgiPlug (BD), and 1:200 CD28/CD49d (FastImmune; BD). Cells were washed with FACS buffer and stained with the indicated cell surface antibodies for 30 minutes at 4°C.  Cells were washed, fixed, and stained with anti-human IFNg Alexa 647 and anti-human TNFα Alexa 488 in permeabilization buffer (BD).  Cells were then washed and collected on an LSRFortessa.<br><br>For Jurkat assays: TCR-expressing Jurkats  were stimulated for 24 hours with HLA allele transduced APCs loaded with 10 µg/ml peptide or vehicle control.  Antigen-reactive CD8+ cells were identified by co-expression of NFAT-mCherry and anti-human CD69 PE. |
| Instrument | BD LSRFortessa |
| Software | BD FACSDiva v9.0 was used for sample collection.<br>Flowjo v10.10.0 was used for analysis. |
| Cell population abundance | No sorting was performed in this study. |
| Gating strategy | Lymphocytes were identified by FSC-A/SSC-A followed by singlet gating using FSC-H/FSC-W.  CD8+ T cells were selected against CD14/CD16/CD19 dump channel negative cells followed by live cell selection.  After gating on TCR+ expressing CD8+ T cells, antigen-specific CD8+ T cells were identified by pMHC tetramer dual positivity (PE/APC) or intracellular cytokine production (IFNg vs TNFa).  After gating on live CD8+ T cells in TCR-expressing Jurkats, antigen reactivity was determined by dual positivity of NFAT-mCherrry and CD69 PE. |

☒ Tick this box to confirm that a figure exemplifying the gating strategy is provided in the Supplementary Information.

