## [Peer Review File · Nature Immunology]

Antigen specificity of clonally-enriched CD8+ T cells in multiple sclerosis

Corresponding Author: Dr Joseph Sabatino, Jr.

Version 0:

Decision Letter:

14th Apr 2025

Dear Joe,

Thank you for providing a point-by-point response to the referees' comments on your manuscript entitled, "Antigen specificity of clonally-enriched CD8+ T cells in multiple sclerosis". As noted previously, while they find your work of considerable potential interest, they have raised quite substantial concerns that must be addressed. In light of these comments, we cannot accept the current manuscript for publication, but would be very interested in considering a revised version that addresses these serious concerns along the lines proposed in your point-by-point rebuttal.

We invite you to submit a substantially revised manuscript, however please bear in mind that we will be reluctant to approach the referees again in the absence of major revisions.

Specifically, the revision should include new experiments to address:

- (1) additional analysis of the antigen specificity of the CSF-expanded CD8+ T cell clones obtained from the MS/CIS patients; in particular if the TCRs of these clones recognize (or not) EBV-specific epitopes
- (2) compare transcriptomics of the two identified EBV-specific CD8+ T cells to other CSF-expanded clonotypes by completing additional differential gene expression analysis (comparing tissue-resident memory T cell subsets)
- (3) test for the presence of EBV RNA transcripts in patient/control CSF supernatants
- (4) test cloned TCRs that were predicted by GLIPH2 analysis for reactivity to EBV epitopes in vitro

Please include the additional textual clarifications and analyses of datasets in hand as indicated in your response letter.

When you revise your manuscript, please take into account all reviewer and editor comments, please highlight all changes in the manuscript text file in Microsoft Word format.

* If you have not done so already please begin to revise your manuscript so that it conforms to our Article format instructions at <http://www.nature.com/ni/authors/index.html>. Refer also to any guidelines provided in this letter.

The Reporting Summary can be found here:

Extended Data figures and tables are online-only (appearing in the online PDF and full-text HTML version of the paper), peer-reviewed display items that provide essential background to the Article but are not included in the printed version of the paper due to space constraints or being of interest only to a few specialists. A maximum of ten Extended Data display items (figures and tables) is typically permitted. When re-submitting your manuscript, please ensure that any supplementary figures and tables that are more critical to the manuscript's conclusions are converted to Extended data to increase these data's visibility.

Link Redacted

If you wish to submit a suitably revised manuscript we would hope to receive it within 6 months. If you cannot send it within this time, please let us know. We will be happy to consider your revision so long as nothing similar has been accepted for publication at Nature Immunology or published elsewhere.

Nature Immunology is committed to improving transparency in authorship. As part of our efforts in this direction, we are now requesting that all authors identified as 'corresponding author' on published papers create and link their Open Researcher and Contributor Identifier (ORCID) with their account on the Manuscript Tracking System (MTS), prior to acceptance. ORCID helps the scientific community achieve unambiguous attribution of all scholarly contributions. You can create and link your ORCID from the home page of the MTS by clicking on 'Modify my Springer Nature account'. For more information please visit www.springernature.com/orcid.

Thank you for the opportunity to review your work.

Kind regards,

Laurie

Laurie A. Dempsey, Ph.D.
Senior Editor
Nature Immunology
l.dempsey@us.nature.com
ORCID: 0000-0002-3304-796X

Referee expertise:

Referee #1: Immune responses to herpesvirus infections

Referee #2: TCR repertoire analysis and T effector cell functions

Referee #3: Neurodegenerative diseases

Reviewers' Comments:

Reviewer #1 (Remarks to the Author):

Manuscript Nr: NI-A39647

The authors demonstrate that T cells in the cerebrospinal fluid (CSF) of patients with multiple sclerosis (MS) or clinically isolated syndrome (CIS as a first clinical MS manifestation) differ from blood in activation and tissue homing markers. Interestingly, clonotype expansions are mostly private to the two tissue compartments and to individual patients with only 1.5% of clonotype overlap between CSF and blood even within the same patient. MS/CIS patients seemed to have slightly larger clonotype expansions in the CSF than other neurological diseases, and the expanded clones made up 20% of all CD8+ T cells in the CSF of one patient. Highly expanded CD8+ T cell clonotypes expressed markers of tissue residency (CD69, CD103[IGTAE]) and KLRG1, KLF2, CCL5, DUSP2, CXCR3 as well as CD27. Of the 23 most expanded TCRs, the authors manage to clone 15 and for 3 of these they were able to identify by phage display peptides that stained them when loaded onto the respective HLA class I tetramers. However, only one TCR enabled transgenic T cells to produce cytokines in response to the cognate peptide. No physiological source of this peptide was identified. TCR antigen prediction then was able to identify Epstein Barr virus (EBV) derived epitopes for two additional TCRs. In a tour de force 98 additional HLA class I tetramers with immunodominant viral peptides (EBV, CMV, influenza, measles, HSV, VZV, HBV, HIV and HTLV1) were tested but no additional specificities were identified. No cross-reactivity of the two EBV specific TCRs towards a panel of autoantigen derived peptides was detected, and around half of CIS/MS patients as well as controls had detectable EBV DNA in their CSF, with more sensitive EBER2 specific ddPCR even nearly all tested CSFs with the highest levels in CIS/MS patients. From these data the authors conclude that EBV infected cells might cause EBV specific CD8+ T cell expansion in the central nervous system (CNS) of MS patients.

These are interesting findings but the antigen specificity of the majority of expanded TCR clonotypes in the studied CIS/MS patients remains unknown. Moreover, additional information should be provided on how representative the two EBV specific T cell clones are for the overall phenotype of CSF expanded clonotypes.

Major comments:

1. The authors were able to increase their sensitivity to detect EBV DNA in CSF by using ddPCR for EBER2. Did they also detect EBV transcripts in CSF B cells in their ssRNAseq dataset? If yes, which transcripts and in which B cell phenotype?
2. The authors report CD69 up-regulation on T cells from MS/CIS patients. Do they also find other markers that are indicative of tissue resident memory T cells? The same applies for the highly expanded CD8+ T cell clones for which they find CD103 (IGTAE) expression in addition. Curiously they observe also KLF2 expression, which is usually down-regulated upon tissue residency. Are KLF2+ and CD69+CD103+ T cells separate cells or even clones?
3. They observe little overlap between TCR clonotype expansion in blood and CSF. Is the blood CD8+ T cell clonotype expansion CMV driven, while the CSF clonotype expansion might at least in part be EBV driven?
4. In some studies, it was argued that early in MS, in CIS patients, EBV specific CD8+ T cells are expanded. Do the authors also observe an increased clonotype expansion in the CIS compared to the MS patients? Considering the CIS patient numbers this is probably difficult to judge but any information would be interesting.
5. The significant CD27 expression of highly expanded CD8+ T cells in the CSF of CIS/MS patients is interesting because it is a hallmark of EBV specific CD8+ T cells (Schmidt et al., Cell Reports 2023). Is CD27 expression particularly high in TCR clonotypes that the authors identified to recognize EBV antigens?
6. Are the two TCRs for which EBV reactivity was identified (EBNA3A and BZLF1) in the respective CIS/MS patients enriched in CSF over blood? Are there other HLA-B*0801 and HLA-B*3501 positive CIS/MS patients in the studied cohort, in which similar TCRs were not enriched in the CSF?
7. How representative are the two EBV specific CD8+ T cell clones of the phenotype of clonally expanded CD8+ T cells in the CSF with respect to tissue residency markers, chemokine production, KLF2 and CD27 expression?

Minor comments:

1. Of the 23 most expanded TCR clonotypes in the CSF of the studied CIS/MS patients, only for 5 TCRs peptides were identified that led to HLA class I tetramer straining and only 3 that elicited T cell responses by cytokine production. I think this should be reflected somewhere in the abstract because it seems difficult to draw general conclusions from these three, nearly anecdotal TCR specificities.

Reviewer #2 (Remarks to the Author):

The manuscript describes the analysis of a cohort of subjects with various types of MS or healthy controls. ScRNA-seq and TCR-Seq was performed on peripheral blood samples and CSF. Transcriptional analysis of the cell states is described, TCR diversity and clonality are reported. Clonally expanded TCRs were cloned for subsequent antigen identification. Two approaches were used—a synthetic mimotope library via yeast display, and literature matching. The yeast display approach nominates several candidates, only one of which is functionally confirmed. The literature matching approach finds two EBV candidate TCRs which are confirmed. Thus, from two subjects, an EBV-specific response in the CSF was identified. The study adds to the growing literature on the relationship between MS and EBV infection and reactivation. The weaknesses include (as the authors note), the relatively small cohort size, and the ultimately only two TCRs that are identified with antigen targets. These TCRs were both known EBV targets.

- 1) The authors report the mimotope identification in the abstract and manuscript, but the significance of this (given the functional assay failures) is unclear. Do they think these are real IDs? Are there any structurally similar peptides (maybe from EBV?) that these might be mirroring? If not it's not clear to me what it adds to the manuscript. As a minor point, on line 225 the authors state that none of the mimotopes were genuine target antigens but they did identify one on line 216.

- 2) Similarly could they screen additional TCRs from the collection against EBV? Maybe EBV infections in patient APCs?
- 3) The authors do a clonotype similarity analysis using GLIPH2. From my understanding of the methods, they ran the entire CSF data set, and then extracted only the expanded clones for plotting in Figure 3b. These results are not revisited after the EBV target ID—were any of those TCRs in clusters? Testing the other connected TCRs would be useful. Also, if I understood the GLIPH2 analysis correctly, they only clustered on TCRbeta—seems like clustering on alpha-beta might be useful (or even just a separate alpha analysis)?

Reviewer #3 (Remarks to the Author):

Mittl and coworkers undertook an analysis, primarily focused on CSF CD8+ T cells coming from research participants with multiple sclerosis (MS). Using a small convenience cohort and a very small number of healthy controls (HC) and other neuroinflammatory disease (OND), they performed single cell RNA-Seq on cells from both blood and CSF. They identified clonally expanded populations of CD8+ T cells in subjects from both MS and HC. Using TCR-Seq data, they undertook to identify shared clonotypes across patients (there were virtually none) and to specify the antigenic targets of the CD8+ T cells. This aspect of the work was executed with admirable diligence and rigor, including unbiased screening of a peptide library as well as challenge with putative viral and self antigens. Following preliminary identification of novel mimotopes from the peptide library and candidate viral antigens, they used two technologies to qualify further their results: staining cells with p-MHC tetramers and introduction of TCRs into primary T cells for verification of functional reactivity. The data converged on a limited number of TCRs recognizing determinants from EBV, which is necessary but not sufficient for MS pathogenesis. Interestingly, the clones recognized epitopes from late-latent and lytic stages of infection. Importantly, HC also contained expanded CD8+ clones, although less numerous and large than those detected in MS subjects' cells.

The primary concern with the manuscript lies with interpretation and discussion with the data. Specific concerns are as follows:

1. "For instance, CD8+ T cells
94 (Fig. 1E, Table S3) in the CSF displayed significantly increased expression of various
95 genes relative to the peripheral blood, including genes associated with migration and
96 trafficking (CXCR3, CXCR4, CCL4, ITGB1, ITGA4), signaling and activation (CD2, FYN,
97 DUSP2), and cytotoxicity (GZMK, GZMA). In contrast, peripheral blood CD8+ T cells
98 expressed significantly higher levels of FOS, JUN, DUSP1, and GADD45B, indicating an
99 alternate activation state."

Comment:

Direct comparison between blood versus CSF CD8+ T cells is not appropriate, and does not support the conclusions stated here. CSF CD8+ T cells are virtually all memory cells while blood is a 50/50 mixture of memory and naive. Therefore simple comparison of blood and CSF will illustrate differences between naive and memory cells. It would be more informative to compare the blood versus CSF memory CD8+ compartment.

2. "115 CCR7, SELL"

Comment: These differences are statistically significant but not likely to be biologically meaningful. Further, the expression of CCR7 and SELL in CSF T cells indicates central memory status, not naive state.

3. "These
129 findings suggest a diverse array of T cell clonotypes may be preferentially recruited in
130 both the blood and CSF of MS/CIS patients."

Comment: The meaning of this sentence is not clear. Recruited from where? Do the authors suggest that, in MS/CIS, the CD8+ clones are concentrated in CSF while such is not the case for CD4+ compartment?

4. "enriched in the CSF relative to the peripheral
148 blood were likely to be responsive to CNS antigens."

comment: It's not certain that these expanded clones are responding to antigens found only or predominantly in CNS, further supported by the lack of reactivity to CNS self-antigens. According to understanding of CNS immune privilege, expanded CSF clones will have been primed in a peripheral immune site and re-stimulated within CNS compartment, likely by meningeal APCs. Blood cells, which are in transit between marrow, lymphoid organs and tissues, are not directly comparable to CSF cells, as noted above. It would be necessary to evaluate another population of tissue resident CD8+ memory cells to know whether these cells reported in this study are authentically enriched in CSF.

5. "Our
338 findings therefore provide further support that EBV may be related to multiple forms of
339 CNS pathology."

Comment: Isn't this finding equally suggestive that, although EBV is unequivocally associated with MS by abundant

seroepidemiologic data, the expanded CD8+ EBV specific clones are passengers, unrelated to pathology?

6. "Alternatively, the findings of CD8+ 348 T cells reactive against EBV late latent and lytic antigens are consistent with other 349 reports^{4,41,45} and could indicate EBV reactivation within the CNS."

Comment: EBV reactivation within CNS is not the only explanation of this finding -- peripherally-activated cells could preferentially traffic to the CNS, for example. Further, EBV reactivation is not a stable, unchanging state but one point in a dynamic cycle of lytic and latent infection (see PMID: 39865738 for MS-relevant discussion). The most direct test of this hypothesis, Atara's phase 2 EMBOLD trial produced a negative result (<https://www.neurologylive.com/view/ata188-fails-meet-primary-end-point-phase-2-embold-study-progressive-ms>) and would need to be discussed in this context.

7. "The detection of 350 EBV DNA in the CSF of most MS patients, including those with confirmed expanded EBV351 specific CD8+ T cells, is strongly supportive of this possibility."

Comment: This statement seems overly forceful, given the equivalent frequency of detecting EBV DNA in HC and MS/CIS. Would it be appropriate to assay for CSF DNA from other Herpesviruses including CMV and HHV6?

8. "The detection 364 of EBV in the CSF of non-MS individuals likely reflects the fact that the majority of the 365 general population, with or without MS, is chronically infected with EBV."

Comment: It would be important to say more about this point: HC do not have B cell rich meningeal aggregates, while many MS patients do have these (they are the source of OCBs), and would strongly bias towards finding EBV DNA in the CSF of MS/CIS. Therefore, unless there is an alternative hypothesis, finding EBV DNA in the CSF of both MS/CIS and HC is likely to be an incidental finding. The trafficking B cell population in HC CSF will be vanishingly small, and could not account for the presence of EBV DNA in CSF. The data are consistent with the possibility that cell-free EBV DNA enters the CSF from blood of both MS research participants and healthy controls.

9. "Pseudotime analysis revealed distinct populations of T cells largely segregated based on 91 compartment (i.e. CSF or blood)"

Is it possible to apply pseudotime analysis across two such different samples as blood and CSF? This approach is typically applied to subpopulations of a single population of cells.

10. "153 More than 70% of the highly expanded and CSF-enriched T cell clonotypes in the CSF were CD8+ 154 T cells."

This observation is impressive given that CD4+ cells in CSF outnumber CD8+ cells more than 2:1 and that the authors sequenced twice as many CD4s as CD8s. This bias towards CD8+ cell clonality deserves further comment.

Version 1:

Decision Letter:

Our ref: NI-A39647A

7th Nov 2025

Dear Dr. Sabatino,

Thank you for submitting your revised manuscript "Antigen specificity of clonally-enriched CD8+ T cells in multiple sclerosis" (NI-A39647A). It has now been seen by two of the original referees and their comments are below. The reviewers find that the paper has improved in revision, and therefore we'll be happy in principle to publish it in Nature Immunology, pending minor revisions to satisfy the referees' final requests and to comply with our editorial and formatting guidelines.

We will now perform detailed checks on your paper and will send you a checklist detailing our editorial and formatting requirements in about a week. Please do not upload the final materials and make any revisions until you receive this additional information from us.

If you had not uploaded a Word file for the current version of the manuscript, we will need one before beginning the editing process; please email that to immunology@us.nature.com at your earliest convenience.

Thank you again for your interest in Nature Immunology Please do not hesitate to contact me if you have any questions.

Kind regards,

Laurie

Laurie A. Dempsey, Ph.D.
Senior Editor
Nature Immunology
l.dempsey@us.nature.com
ORCID: 0000-0002-3304-796X

Reviewer #1 (Remarks to the Author):

Manuscript Nr: NI-A39647A
Hayashi, Mittl et al., "Antigen specificity of clonally-enriched CD8+ T cells in multiple sclerosis"

The authors demonstrate that T cells in the cerebrospinal fluid (CSF) of patients with multiple sclerosis (MS) or clinically isolated syndrome (CIS as a first clinical MS manifestation) differ from blood in activation and tissue homing markers. Interestingly, clonotype expansions are mostly private to the two tissue compartments and to individual patients with only 1.5% of clonotype overlap between CSF and blood even within the same patient. MS/CIS patients seemed to have slightly larger clonotype expansions in the CSF than other neurological diseases, and the expanded clones made up 20% of all CD8+ T cells in the CSF of one patient. Highly expanded CD8+ T cell clonotypes expressed markers of tissue residency (CD69, CD103[IGTAE]) and KLRG1, KLF2, CCL5, DUSP2, CXCR3 as well as GD27. Of the 23 most expanded TCRs, the authors manage to clone 15 and for 3 of these they were able to identify by phage display peptides that stained them when loaded onto the respective HLA class I tetramers. However, only one TCR enabled transgenic T cells to produce cytokines in response to the cognate peptide. No physiological source of this peptide was identified. TCR antigen prediction then was able to identify Epstein Barr virus (EBV) derived epitopes for two additional TCRs. In a tour de force 98 additional HLA class I tetramers with immunodominant viral peptides (EBV, CMV, influenza, measles, HSV, VZV, HBV, HIV and HTLV1) were tested but no additional specificities were identified. No cross-reactivity of the two EBV specific TCRs towards a panel of autoantigen derived peptides was detected, and around half of CIS/MS patients as well as controls had detectable EBV DNA in their CSF, with more sensitive EBER2 specific ddPCR even nearly all tested CSFs with the highest levels in CIS/MS patients. From these data the authors conclude that EBV infected cells might cause EBV specific CD8+ T cell expansion in the central nervous system (CNS) of MS patients.

In their revised manuscript version, the authors have addressed all of my concerns, even so some could not be clarified, such as major concern #4, due to the studied patient population. Nevertheless, they could show that BamH1 W transcripts but not EBER2, EBNA3A and BZLF1 mRNAs were elevated along with EBV DNA in a subset of CSFs. They also now clarified that while most expanded T cell clonotypes had a Trm phenotype (CD69+CD103+KLF2low) the identified EBV specific T cell clonotypes were rather CXCR5+CD27+, germinal center homing and possibly protective T cell responses against EBV infected B cells. Finally, they could predict low EBV and CMV specific T cell clonotype numbers, based on TCR specificity prediction, in peripheral blood, while CMV specific T cell clonotypes seemed to be absent in CSF and EBV possibly enriched. Therefore, I find the manuscript improved.

Minor comment:

1. In the revision period the authors identified a third TCR that recognized HLA-B*2902 matched EBV transformed B cells (LCLs). So far, the authors have not been able to assign an EBV peptide specificity to this clone but surprisingly they do not seem to have tested the two published HLA-A29 restricted CD8+ T cell epitopes EBNA3A aa491-499 VFSDGRVAC and LMP2 aa349-358 ILLARLFY. Testing these could be considered to extend the conclusions at least to 3 EBV specificities, if one of these published epitopes is recognized by the third TCR.

Reviewer #2 (Remarks to the Author):

The authors have undertaken several experiments to expand their specificity assignments and were able to confirm (by my count) 2 more EBV targets. The numbers are still sparse, but this is an incredibly difficult task with current approaches (needles in a nearly infinite haystack) so I think the fact that they were able to identify the new A29 specificity is remarkable. Given the value of the human samples and subsequent analysis, I have no further concerns.

Version 2:

Decision Letter:

In reply please quote: NI-A39647B

Dear Joe,

I am delighted to accept your manuscript entitled "Antigen specificity of clonally-enriched CD8+ T cells in multiple sclerosis"

for publication in an upcoming issue of Nature Immunology.

Over the next few weeks, your paper will be copyedited to ensure that it conforms to Nature Immunology style. Once your paper is typeset, you will receive an email with a link to choose the appropriate publishing options for your paper and our Author Services team will be in touch regarding any additional information that may be required.

Authors may need to take specific actions to achieve compliance with funder and institutional open access mandates. If your research is supported by a funder that requires immediate open access (e.g. according to [Plan S principles](https://www.springernature.com/gp/open-science/plan-s-compliance) or the [NIH public access policy](https://www.springernature.com/gp/open-science/us-federal-agency-compliance)) then you should select the gold OA route, and we will direct you to the compliant route where possible. Because authors warrant under our subscription licensing terms that they haven't committed to licensing any version of their article under a licence inconsistent with the terms of our agreement – including the applicable embargo period – publication under the subscription model isn't suitable for authors whose funders require no embargo.

Your paper will be published online soon after we receive your corrections and will appear in print in the next available issue.

Also, if you have any spectacular or outstanding figures or graphics associated with your manuscript - though not necessarily included with your submission - we'd be delighted to consider them as candidates for our cover. Simply send an electronic version (accompanied by a hard copy) to us with a possible cover caption enclosed.

If you have not already done so, we strongly recommend that you upload the step-by-step protocols used in this manuscript to protocols.io. protocols.io is an open online resource that allows researchers to share their detailed experimental know-how. All uploaded protocols are made freely available and are assigned DOIs for ease of citation. Protocols can be linked to any publications in which they are used and will be linked to from your article. You can also establish a dedicated workspace to collect all your lab Protocols. By uploading your Protocols to protocols.io, you are enabling researchers to more readily reproduce or adapt the methodology you use, as well as increasing the visibility of your protocols and papers. Upload your Protocols at <https://protocols.io>. Further information can be found at <https://www.protocols.io/help/publish-articles>.

Please note that we encourage the authors to self-archive their manuscript (the accepted version before copy editing) in their institutional repository, and in their funders' archives, six months after publication. Nature Portfolio recognizes the efforts of funding bodies to increase access of the research they fund, and strongly encourages authors to participate in such efforts. For information about our editorial policy, including license agreement and author copyright, please visit www.nature.com/ni/about/ed_policies/index.html

Best wishes for the Holidays

Laurie

Laurie A. Dempsey, Ph.D.
Senior Editor
Nature Immunology
l.dempsey@us.nature.com
ORCID: 0000-0002-3304-796X

Click here if you would like to recommend Nature Immunology to your librarian
<http://www.nature.com/subscriptions/recommend.html#forms>

** Visit the Springer Nature Editorial and Publishing website at http://editorial-jobs.springernature.com?utm_source=ejp_NImm_email&utm_medium=ejp_NImm_email&utm_campaign=ejp_NImm for more information about our career opportunities. If you have any questions please click [here](mailto:editorial.publishing.jobs@springernature.com).

We would like to thank all the reviewers for their very thoughtful and constructive feedback. We have considered all comments and questions seriously, which we have attempted to fully address in our revised manuscript. We have undertaken a number of new experiments, including studies to further address EBV CD8+ T cell specificity and EBV gene expression in CSF. We have also performed additional transcriptomic and clonal analyses to further delineate the phenotype of EBV-specific CD8+ T cells. Finally, we have made a number of modifications and additions to the interpretation of the findings from our study. A number of additional co-authors were included as a result of the new data provided. All tracked changes were included in the revised manuscript with a cleaned version of the manuscript also included.

We have provided a point-by-point response to each of the Reviewers' comments and questions below. Where applicable, new data is shown after each response with a description of the new figure or table corresponding to its location in the revised manuscript. We hope the reviewers find our responses satisfactory and thank them again for their willingness to reconsider our manuscript.

Reviewer #1

(Remarks to the Author)

Manuscript Nr: NI-A39647

Mittl et al., "Antigen 1 specificity of clonally-enriched CD8+ T cells in multiple sclerosis"

The authors demonstrate that T cells in the cerebrospinal fluid (CSF) of patients with multiple sclerosis (MS) or clinically isolated syndrome (CIS as a first clinical MS manifestation) differ from blood in activation and tissue homing markers. Interestingly, clonotype expansions are mostly private to the two tissue compartments and to individual patients with only 1.5% of clonotype overlap between CSF and blood even within the same patient. MS/CIS patients seemed to have slightly larger clonotype expansions in the CSF than other neurological diseases, and the expanded clones made up 20% of all CD8+ T cells in the CSF of one patient. Highly expanded CD8+ T cell clonotypes expressed markers of tissue residency (CD69, CD103[IGTAE]) and KLRG1, KLF2, CCL5, DUSP2, CXCR3 as well as CD27. Of the 23 most expanded TCRs, the authors manage to clone 15 and for 3 of these they were able to identify by phage display peptides that stained them when loaded onto the respective HLA class I tetramers. However, only one TCR enabled transgenic T cells to produce cytokines in response to the cognate peptide. No physiological source of this peptide was identified. TCR antigen prediction then was able to identify Epstein Barr virus (EBV) derived epitopes for two additional TCRs. In a tour de force 98 additional HLA class I tetramers with immunodominant viral peptides (EBV, CMV, influenza, measles, HSV, VZV, HBV, HIV and HTLV1) were tested but no additional specificities were identified. No cross-reactivity of the two EBV specific TCRs towards a panel of autoantigen derived peptides was detected, and around half of CIS/MS patients as well as controls had detectable EBV DNA in their CSF, with more sensitive EBER2 specific ddPCR even nearly all tested CSFs with the highest levels in CIS/MS patients. From these data the authors conclude that EBV infected cells might cause EBV specific CD8+ T cell expansion in the central nervous system (CNS) of MS patients.

These are interesting findings but the antigen specificity of the majority of expanded TCR clonotypes in the studied CIS/MS patients remains unknown.

Response: This is a very important point, and we have included additional language in the manuscript to acknowledge this. We have also performed a number of additional experiments to further investigate the EBV-specificity of additional CSF-expanded CD8+ T cells. Given the

inherent challenges in querying a wide array of virus epitopes and HLA restriction considerations, we opted to use EBV-transformed lymphoblastoid cell lines (LCLs) as a means of testing EBV-specificity of candidate CSF-expanded CD8+ TCRs. Focusing on TCRs that did not have any known antigen specificity, we co-cultured Jurkat reporter cells expressing a single candidate CD8+ TCR with partially HLA-matched LCLs (we opted for partial rather than full HLA match in order to determine the potential MHC allele restriction). Fully HLA-mismatched LCLs and mismatched TCR-expressing Jurkats were used as controls. In this way, we were able to test 16 candidate TCRs against at least 2 different LCLs matching at 3-6 MHC I alleles (**Table S14**). While nearly all TCRs showed no detectable response, TCR 94669_8198 demonstrated a clear, reproducible response to LCLs only when matching the HLA-A*29:02 allele (**Fig. 6A-C**). No response was elicited from primary B cells from the same donor used to generate the LCLs, indicating this is very likely an EBV-specific response rather than a B cell self-antigen or alloreactive response (**Fig. 6B**). This EBV-specific CD8+ TCR is from a third MS patient, MS27. The corresponding new text is found in lines 294-307 of the manuscript.

Figure 6. CSF-expanded CD8+ T cell reactivity to EBV-transformed B cells. Representative flow cytometry analysis of Jurkat reporter cells expressing the indicated TCR and CD8 co-receptor that were co-cultured with partially HLA-matched (matching allele indicated in red) EBV-transformed lymphoblastoid cell lines (LCLs). Reactivity was assessed by co-expression of CD69 and NFAT-mCherry. LCLs that were fully HLA-mismatched and mismatched TCR-expressing Jurkat cells were used as negative controls (A). Representative flow cytometry analysis of Jurkat reporter cells expressing TCR 94669_8198 were co-cultured with LCLs or primary uninfected B cells from the same donor (B). Summary of all candidate TCRs tested and the corresponding matching MHC I alleles expressed by different LCL lines is

shown in panel C. The mCherry/CD69 signal of a given TCR-expressing Jurkat co-cultured with partially MHC-matched LCLs was normalized to the signal observed from completely MHC-mismatched LCLs (left bar graph) or mismatched TCR-expressing Jurkats (right bar graph), which was reported as fold-change (FC).

In order to identify the potential specific epitope, we created an HLA-A*29:02-expressing cell line and tested reactivity to 4 candidate EBV peptides identified from IEDB (FLYALALLL, VFGQQAYFY, AYSSWMYSY, FVYGGSKTSLY). However, no response was elicited (**Fig. S9**), indicating reactivity to a still unspecified EBV epitope. The corresponding new text is found in lines 307-312 of the manuscript.

Figure S9. A*29:02 EBV peptide testing. Jurkat reporter cells expressing TCR 94669_8198 were co-cultured with HLA-A*29:02-expressing K562 cells pulsed with the indicated EBV peptides or no peptide for 24 hours. Antigen reactivity was assessed by co-expression of CD69 and NFAT-mCherry.

Major comments:

1. The authors were able to increase their sensitivity to detect EBV DNA in CSF by using ddPCR for EBER2. Did they also detect EBV transcripts in CSF B cells in their ssRNAseq dataset? If yes, which transcripts and in which B cell phenotype?

Response: The reviewer poses a very important question. This is very challenging to address directly due to several technical limitations. In addition to low numbers of B cells in CSF, the short-read nature of scRNA-seq prevents identification of transcripts that can be confidently aligned to EBV or other non-human sequences. Indeed, a recent study of scRNA-seq from CSF of MS patients failed to identify any clear enrichment of viral genomes¹. Furthermore, we have already performed analysis of bulk RNA-seq from CSF B cells of the same cohort of patients and controls, which did not reveal any clear EBV-specific transcripts down to a limit of detection of 10 pg RNA (equivalent to one cell's RNA)². Another limitation is that a number of key EBV transcripts are non-coding (including EBER1/2), and thus not captured by polyA selection in single cell sequencing.

To address this question another way, we have performed ddPCR of select EBV RNA transcripts from cell-free CSF supernatants from the same patients in this study. Using random primers for cDNA generation, we were able to test for coding and non-coding RNA transcripts corresponding to latent and lytic genes, including *EBER2*, *BamHI-W* repeat region (contains promoter for multiple EBNA genes), *EBNA3A*, and *BZLF1*. Not surprisingly, *EBER2* cDNA was overall less detectable than DNA and there was no significant difference between CIS/MS and HC/OND (**Fig. 7A-B**). *EBNA3A* (latency III gene) and *BZLF1* (early lytic gene) were mostly undetectable with no significant difference between the two cohorts (**Fig. 7B**). Strikingly, we did observe a significant increase in *BamHI-W* transcripts in CIS/MS patients compared to controls

(Fig. 7B), including the 3 MS patients with highly expanded EBV-specific CD8+ T cells in CSF. This therefore suggests that EBV reactivation is enhanced in CIS/MS, which drives expansion of EBV-specific CD8+ T cells. The corresponding new text is found in lines 379-387 of the manuscript.

Figure 7. Detection of EBV DNA and transcripts in CSF. Summary of EBV DNA ddPCR results from CSF supernatant in which *EBER2* was normalized to a housekeeping gene (A). EBV cDNA for each the indicated genes were measured by ddPCR and normalized to a housekeeping gene (B). Each sample was run in duplicate, and each dot represents the average result from each subject (mean and SEM shown). Note: n=4 in HC/OND for *BamHI-W* due to lack of sufficient sample for OND4.

2. The authors report CD69 up-regulation on T cells from MS/CIS patients. Do they also find other markers that are indicative of tissue resident memory T cells? The same applies for the highly expanded CD8+ T cell clones for which they find CD103 (ITGAE) expression in addition. Curiously they observe also KLF2 expression, which is usually down-regulated upon tissue residency. Are KLF2+ and CD69+CD103+ T cells separate cells or even clones?

Response: We have performed a comprehensive transcriptomic analysis of CSF-expanded T cells comparing Trm positive (defined by co-expression of CD69 and ITGAE) and Trm negative cells (Fig. S5A, Table S9). In support of the reviewer's question, *KLF2* expression is minimally expressed in Trm cells and is significantly higher in Trm negative versus Trm positive CSF-expanded T cells (Fig. S5B). The corresponding new text is found in lines 201-203 of the manuscript.

Figure S5. Tissue resident memory T cells in CSF. Expression levels and percent expression of the indicated genes is shown for tissue resident memory (Trm) positive (CD69+ and ITGAE+) and Trm negative CD4+ and CD8+ T cells (A). KLF2 expression levels between Trm negative and positive T cells (B). * $p < 0.05$.

3. They observe little overlap between TCR clonotype expansion in blood and CSF. Is the blood CD8+ T cell clonotype expansion CMV driven, while the CSF clonotype expansion might at least in part be EBV driven?

Response: This is a really interesting question, in particular given that CSF enrichment of CD8+ T cells specific for viral antigens other than EBV have not been described in MS patients³⁻⁵. Because it is not possible to test every single TCR from the CSF and blood individually for CMV and EBV specificity, we have attempted to address this question by performing TCR sequencing alignment (matching V and J genes and CDR3 amino acids of paired TCR $\alpha\beta$ chains) of all expanded CD8+ TCR sequences (> 1 TCR) from the blood and CSF of all patients and controls against EBV and CMV sequences from the VDJdb database. We observed identical, relatively low frequencies (0.14%) of EBV- and CMV-aligned CD8+ TCR sequences in the blood (**Fig. 5E**). In contrast, no CSF TCR sequences aligned to CMV while 1.39% aligned to EBV. We further subset the EBV-aligned CD8+ TCR sequences between MS/CIS and HC/OND patients. Strikingly, EBV-specific TCR alignment was only observed in MS/CIS patients but not in HC/OND (**Fig. 5F**). This provides support that EBV preferentially drives CD8+ T cell expansion in the CSF, but not the blood with no clear impact from CMV. The corresponding new text is found in lines 332-342 of the manuscript.

Figure 5. EBV reactivity of highly expanded CSF-enriched CD8+ T cells. TCR sequencing alignment of expanded CD8+ TCRs to EBV- and CMV-specific TCRs in the blood (PB) and CSF (E). EBV-specific TCR alignment of all expanded CD8+ T cell clonotypes PB and CSF of MS/CIS and HC/OND subjects (F).

4. In some studies, it was argued that early in MS, in CIS patients, EBV specific CD8+ T cells are expanded. Do the authors also observe an increased clonotype expansion in the CIS compared to the MS patients? Considering the CIS patient numbers this is probably difficult to judge but any information would be interesting.

Response: We agree this is an interesting question. The proportions of non-expanded (singleton TCRs), moderately expanded (≥ 1 TCR but $< 0.75\%$) and highly expanded ($\geq 0.75\%$) CD4+ and CD8+ TCRs in the CSF of CIS and MS patients were quite similar as shown in **Table S6** (lines 165-166 of the manuscript). Given the limited number of CIS patients in our cohort, it is difficult to make any definite conclusions as the reviewer pointed out. Speaking to the reviewer’s overall question regarding differences in TCR expansion in early versus later MS, our MS cohort is heavily skewed towards early onset as the average disease duration is 8 months relative to the time of sample collection. In future studies, it will be informative to address this question with a more diverse cohort of MS patients, including those measured longitudinally. We have incorporated this into the discussion (lines 479-480 of the manuscript).

	CIS CD4	MS CD4	CIS CD8	MS CD8
Non-expanded	83.9%	85.7%	49.5%	58.4%
Moderately expanded	15.7%	12.5%	39.6%	31.0%
Highly expanded	0.4%	1.9%	10.9%	10.6%

Table S6. Comparison of CD4+ and CD8+ T cell expansion in CSF between CIS and MS patients.

The significant CD27 expression of highly expanded CD8+ T cells in the CSF of CIS/MS patients is interesting because it is a hallmark of EBV specific CD8+ T cells (Schmidt et al., Cell Reports 2023). Is CD27 expression particularly high in TCR clonotypes that the authors identified to recognize EBV antigens?

How representative are the two EBV specific CD8+ T cell clones of the phenotype of clonally expanded CD8+ T cells in the CSF with respect to tissue residency markers, chemokine production, KLF2 and CD27 expression?

Moreover, additional information should be provided on how representative the two EBV specific T cell clones are for the overall phenotype of CSF expanded clonotypes.

Response: The reviewer raises very interesting questions. We have compared the gene expression profiles of the three EBV-specific CD8+ T cell clonotypes to the other CSF-expanded clonotypes in several ways. Differential gene expression analysis identified 3 genes that were significantly increased in the EBV-specific CD8+ T cells (**Table S16**). The most notable was *CXCR5*, which is associated with migration into B cell follicles and control of chronic infections⁶.

We have also compared *CD27* as well as other markers associated with tissue residency, memory differentiation, and chemokine production (*KLF2*, *ITGAE*, *CD69*, *S1PR1*, *CXCR3*, *CXCR4*, *CCL4* and *CCL5*) in the EBV-specific CD8+ T cell clonotypes we identified compared to all other remaining highly expanded CSF-enriched CD8+ T cells. *CD27* was particularly abundant amongst the three pooled EBV-specific CD8+ T cell clonotypes compared to all other highly expanded CD8+ T cell clonotypes (**Fig. S10B**), consistent with the findings by Schmidt et al⁷. In addition, *KLF2*, *CXCR4*, *S1PR1*, and *CCL4* were more abundantly expressed in highly expanded EBV-specific CD8+ T cells. In contrast, *CD69*, *ITGAE*, *CXCR3*, and *CCL5* were more abundant in non-EBV-specific expanded CD8+ T cells. These findings therefore indicate a distinct phenotype of CSF-expanded CD8+ T cells that are specific for EBV. Rather than expressing Trm markers and genes associated with lymphocyte recruitment, these findings suggest that EBV-expanded CD8+ T cells in CSF are an effector population associated with follicular homing and B cell interactions. The corresponding new text is found in lines 343-358 of the manuscript.

Figure S10. EBV-specific CD8+ T cell clonal analysis. The expression of specific genes between the three CSF-expanded EBV-specific CD8+ T cell clonotypes (TCRs 86333_1456, 69317_24418, and 94669_8198 combined) and all other CSF-expanded enriched CD8+ T cells was compared (B).

Are the two TCRs for which EBV reactivity was identified (EBNA3A and BZLF1) in the respective CIS/MS patients enriched in CSF over blood? Are there other HLA-B*0801 and HLA-B*3501 positive CIS/MS patients in the studied cohort, in which similar TCRs were not enriched in the CSF?

Response: Yes, all three of the EBV-specific CD8+ T cell clonotypes identified met our definition of high expansion (> 0.75% of the CSF population) and were enriched at least 2-fold in the CSF relative to the blood as described in the Results and Methods sections (lines 229-230 and 606-607). These clonotypes are quite unique as they are derived from the 23 CD8+ T cell clonotypes that are highlighted in **Fig. 2C**.

The overwhelming majority of T cell clonotypes across all individuals were not enriched in the CSF (all black clonotypes except those on the x-axis of **Fig. 2B**). When considering only the 3 patients with EBV-specific CSF-expanded and enriched CD8+ T cells, only 14 total T cell clonotypes were expanded and enriched in the CSF (red clonotypes below), while the remaining were unenriched (**Fig. S10A**). This suggests that CD8+ T cell expansion and enrichment in the CSF is driven by specific antigen encounter, such as EBV, rather than a more general feature of T cell clonality. The corresponding new text is found in lines 329-332 of the manuscript.

Figure S10. EBV-specific CD8+ T cell clonal analysis. The TCR clonotype frequencies for all clonotypes of the three MS patients (MS6, MS8, MS27) with CSF-expanded EBV-specific CD8+ T cells was compared between the blood and CSF (A). Clonal frequency of T cell clonotypes in the CSF that were highly expanded (at least 0.75%) and enriched at least 2-fold more frequently than the blood of the same individual are highlighted in red.

Minor comments:

1. Of the 23 most expanded TCR clonotypes in the CSF of the studied CIS/MS patients, only for 5 TCRs peptides were identified that led to HLA class I tetramer straining and only 3 that elicited T cell responses by cytokine production. I think this should be reflected somewhere in the abstract because it seems difficult to draw general conclusions from these three, nearly anecdotal TCR specificities.

Response: The reviewer raises a fair point that only a subset of the clonally enriched CD8+ TCRs were found to have some degree of antigen specificity while the remainder of the

assessed TCRs are orphans. We have modified the abstract to clarify this limitation (lines 47 and 51).

Reviewer #2

(Remarks to the Author)

The manuscript describes the analysis of a cohort of subjects with various types of MS or healthy controls. ScRNA-seq and TCR-Seq was performed on peripheral blood samples and CSF. Transcriptional analysis of the cell states is described, TCR diversity and clonality are reported. Clonally expanded TCRs were cloned for subsequent antigen identification. Two approaches were used—a synthetic mimotope library via yeast display, and literature matching. The yeast display approach nominates several candidates, only one of which is functionally confirmed. The literature matching approach finds two EBV candidate TCRs which are confirmed. Thus, from two subjects, an EBV-specific response in the CSF was identified. The study adds to the growing literature on the relationship between MS and EBV infection and reactivation. The weaknesses include (as the authors note), the relatively small cohort size, and the ultimately only two TCRs that are identified with antigen targets. These TCRs were both known EBV targets.

1) The authors report the mimotope identification in the abstract and manuscript, but the significance of this (given the functional assay failures) is unclear. Do they think these are real IDs? Are there any structurally similar peptides (maybe from EBV?) that these might be mirroring? If not it's not clear to me what it adds to the manuscript. As a minor point, on line 225 the authors state that none of the mimotopes were genuine target antigens but they did identify one on line 216.

Response: The reviewer raises an important point. While the unbiased antigen discovery approach revealed only non-naturally mimotopes as candidate antigens, this was an important line of investigation to ensure that our focus on viral specificity was not overly biased and to enable the identification of potential novel MS antigens. Thus, even though the results of yeast display antigen screening did not reveal any definite new antigens, we believe they are important nonetheless as they augment the importance of the EBV findings. Furthermore, the tetramer-binding mimotopes provide a basis for identifying other TCRs with similar specificities, even if the naturally occurring antigen remains unknown.

We also thank the author for highlighting the unclear language regarding “genuine” antigens. We have clarified the language in the cited sections relating to the meaning of mimotopes as well as to the meaning of antigens eliciting functional reactivity. The corresponding text is found in lines 52-54, 251, 256-257, 422-424, and 427-428 of the manuscript.

2) Similarly could they screen additional TCRs from the collection against EBV? Maybe EBV infections in patient APCs?

Response: As described above in response to Reviewer 1, we have completed testing of the other CSF-expanded CD8+ T cells to EBV by co-culturing TCR-expressing CD8+ Jurkats (with NFAT-mCherry reporter) with EBV-infected lymphoblastoid cell lines (LCLs) that are partially HLA-matched (HLA mismatched LCLs and TCR-mismatched T cells as negative controls). Using this approach, we tested 16 additional candidate orphan TCRs against at least 2 different LCLs, ensuring at 3-6 MHC I alleles were covered for each TCR (**Table 14**). Although most TCRs had no measurable reactivity, one TCR, 94669_8198 from patient MS27, showed a

significant response to HLA-A*29:02-expressing LCLs (Fig. 6). The corresponding new text is found in lines 294-307 of the manuscript.

Figure 6. CSF-expanded CD8+ T cell reactivity to EBV-transformed B cells. Representative flow cytometry analysis of Jurkat reporter cells expressing the indicated TCR and CD8 co-receptor that were co-cultured with partially HLA-matched (matching allele indicated in red) EBV-transformed lymphoblastoid cell lines (LCLs). Reactivity was assessed by co-expression of CD69 and NFAT-mCherry. LCLs that were fully HLA-mismatched and

mismatched TCR-expressing Jurkat cells were used as negative controls (A). Representative flow cytometry analysis of Jurkat reporter cells expressing TCR 94669_8198 were co-cultured with LCLs or primary uninfected B cells from the same donor (B). Summary of all candidate TCRs tested and the corresponding matching MHC I alleles expressed by different LCL lines is shown in panel C. The mCherry/CD69 signal of a given TCR-expressing Jurkat co-cultured with partially MHC-matched LCLs was normalized to the signal observed from completely MHC-mismatched LCLs (left bar graph) or mismatched TCR-expressing Jurkats (right bar graph), which was reported as fold-change (FC).

This TCR did not respond to primary B cells from the same donor used to generate the LCLs (**Fig. 6B**) nor to an HLA-A*29:02-expressing APC line pulsed with 4 different EBV peptides identified from IEDB (**Fig. S9**). These experiments therefore indicated that there are 3 confirmed highly CSF-expanded and enriched CD8+ T cell clonotypes from 3 different patients that are EBV-specific, further augmenting the relevance of these clonotypes to MS. The corresponding new text is found in lines 307-312 of the manuscript.

Figure S9. A*29:02 EBV peptide testing. Jurkat reporter cells expressing TCR 94669_8198 were co-cultured with HLA-A29*02-expressing K562 cells pulsed with the indicated EBV peptides or no peptide for 24 hours. Antigen reactivity was assessed by co-expression of CD69 and NFAT-mCherry.

3) The authors do a clonotype similarity analysis using GLIPH2. From my understanding of the methods, they ran the entire CSF data set, and then extracted only the expanded clones for plotting in Figure 3b. These results are not revisited after the EBV target ID—were any of those TCRs in clusters? Testing the other connected TCRs would be useful. Also, if I understood the GLIPH2 analysis correctly, they only clustered on TCRbeta—seems like clustering on alpha-beta might be useful (or even just a separate alpha analysis)?

Response: This is an excellent question. While the question regarding clustering by TCR α is logical, GLIPH⁸ and GLIPH2⁹ were both developed from TCR β sequences only as TCR α sequence motifs were not consistently found for certain antigen determinants. For that reason, further TCR clustering was not performed.

We explored the potential EBV-specificity of all TCR sequences that were identified by our GLIPH2 analysis to be related to the three EBV-specific CD8+ TCR clonotypes. We focused on only GLIPH2-derived TCR sequences from CD8+ T cells that shared the same MHC I allele as that of the aligned EBV-specific clonotype – i.e. TCRs from CD4+ T cells or with no matching MHC I alleles were excluded (**Table S15**). We therefore devoted our efforts on the 2 GLIPH2-aligned TCRs for TCR 86333_1456 and 1 GLIPH2-aligned TCR for TCR 69317_24418 (**Fig. 3C**). The paired TCR α and β chains for each were cloned and expressed in Jurkat reporter cells as previously described. Unlike TCR 86333_1456, the GLIPH2-aligned TCRs 86333_17042 and 86333_18519 showed no detectable reactivity to cognate EBNA3A peptide

FLRGRAYGL (**Fig. 3C, left**). However, TCR 53778_13077, was found by VDJdb to be an exact match to a previously validated EBV-specific TCR¹⁰ reactive to the exact same BZLF1 B*35:01-restricted peptide EPLPQGQLTAY as TCR 69317_24418. We were able to successfully validate this reactivity (**Fig. 3C, right**). TCR 53778_13077 is from patient MS10 and is moderately expanded with a higher frequency in the CSF (0.35%) compared to the blood (0.12%), indicating three-fold enrichment in the CSF. These findings further augment our findings that multiple expanded CD8+ T cell clonotypes that are expanded and enriched in the CSF of MS patients are specific for EBV antigens. The corresponding new text is found in lines 313-326 of the manuscript.

Figure 3. T cell clonal relationships. The indicated GLIPH2-aligned CD8+ TCRs to the EBV-specific TCRs 86333_1456 (left) and 69317_24418 (right) were expressed in reporter Jurkat cells tested for reactivity to the corresponding EBV peptides or no stimulation control (C). FLRGRAYGL (EBV EBNA3A₁₉₃₋₂₀₁) was presented by HLA-B*08:01-expressing APCs (left) and EPLPQGQLTAY (EBV BZLF1₅₄₋₆₄) was presented by HLA-B*35:01-expressing APCs (right).

Reviewer #3

The primary concern with the manuscript lies with interpretation and discussion with the data. Specific concerns are as follows:

"For instance, CD8+ T cells (Fig. 1E, Table S3) in the CSF displayed significantly increased expression of various genes relative to the peripheral blood, including genes associated with migration and trafficking (CXCR3, CXCR4, CCL4, ITGB1, ITGA4), signaling and activation (CD2, FYN, DUSP2), and cytotoxicity (GZMK, GZMA). In contrast, peripheral blood CD8+ T cells expressed significantly higher levels of FOS, JUN, DUSP1, and GADD45B, indicating an alternate activation state."

Comment:

Direct comparison between blood versus CSF CD8+ T cells is not appropriate, and does not support the conclusions stated here. CSF CD8+ T cells are virtually all memory cells while blood is a 50/50 mixture of memory and naive. Therefore simple comparison of blood and CSF will illustrate differences between naive and memory cells. It would be more informative to compare the blood versus CSF memory CD8+ compartment.

Response: The reviewer raises an important point, and we agree that a comparison of memory CD8+ T cells between the compartments would be more informative. We have now performed differential gene expression analysis of memory CD27+ CD8+ T cells between the blood and CSF (**Table S4**). When focusing on genes expressed in at least 10% of memory CD8+ T cells in the blood and/or CSF, there was a relatively small portion of differentially expressed genes (**Fig. S2**). Significant upregulated genes in memory CD8+ T cells in CSF include those associated with T cell activation (*HLA-DRA*), chemokines (*CCL4*, *XCL1*), and cholesterol metabolism (*LDLR*, *SQLE*). Interestingly, significantly downregulated genes in memory CD8+ T cells in the CSF were primarily related to TCR signaling and downstream transcriptional program (*FOS*, *FOSB*, *JUN*, *JUNB*). These findings have been incorporated into the manuscript. The corresponding new text is found in lines 120-124 of the manuscript.

Figure S2. Memory T cell gene expression. Volcano plot analysis of differential gene expression between memory (CD27+) T cells in the CSF and blood. Only genes expressed in at least 10% of memory CD8+ T cells in the blood and/or CSF were analyzed. Genes with adjusted p-values < 0.05 and log₂ fold change > 2 are indicated in red and genes with adjusted p-values < 0.05 in blue.

2. "115 CCR7, SELL"

Comment: These differences are statistically significant but not likely to be biologically meaningful. Further, the expression of CCR7 and SELL in CSF T cells indicates central memory status, not state.

Response: We agree that *CCR7* and *SELL* gene expression in and of themselves do not distinguish naïve from central memory cells, the former being less abundant in CSF. We have modified the language related to the interpretation on these genes in the manuscript (lines 135 and 140).

3. "These findings suggest a diverse array of T cell clonotypes may be preferentially recruited in both the blood and CSF of MS/CIS patients."

Comment: The meaning of this sentence is not clear. Recruited from where? Do the authors suggest that, in MS/CIS, the CD8+ clones are concentrated in CSF while such is not the case for CD4+ compartment?

Response: We apologize for the imprecise wording. Our intention was simply to highlight that a more diverse array of T cell clonotypes (both CD4 and CD8) are present in the blood and CSF of MS/CIS patients. We have revised the wording accordingly in the manuscript (lines 154-155).

4. "enriched in the CSF relative to the peripheral blood were likely to be responsive to CNS antigens."

comment: It's not certain that these expanded clones are responding to antigens found only or predominantly in CNS, further supported by the lack of reactivity to CNS self-antigens. According to understanding of CNS immune privilege, expanded CSF clones will have been primed in a peripheral immune site and re-stimulated within CNS compartment, likely by meningeal APCs. Blood cells, which are in transit between marrow, lymphoid organs and tissues, are not directly comparable to CSF cells, as noted above. It would be necessary to evaluate another population of tissue resident CD8+ memory cells to know whether these cells reported in this study are authentically enriched in CSF.

Response: We agree that our original phrasing was imprecise. The reviewer is correct that CSF enrichment of specific T cell clonotypes does not necessarily indicate reactivity to CNS-specific antigens. We have modified the wording to indicate that that CSF-enriched T cell clonotypes were "more likely to be responsive to local antigens in the CSF and/or CNS (albeit not necessarily CNS-specific antigens)" in lines 174-175 of the manuscript.

5. "Our findings therefore provide further support that EBV may be related to multiple forms of CNS pathology."

Comment: Isn't this finding equally suggestive that, although EBV is unequivocally associated with MS by abundant seroepidemiologic data, the expanded CD8+ EBV specific clones are passengers, unrelated to pathology?

Response: The reviewer raises a very important question. While we contend that EBV expression/reactivation is the likely driver of CD8+ T cell expansion in the CSF, we absolutely agree that our findings do not directly indicate a pathogenic role of EBV and/or EBV-specific CD8+ T cells in MS or any other context. Alternatively, it is possible that EBV reactivation is an epiphenomenon and not directly related to MS pathology¹¹. For instance, memory B cell differentiation into plasma cells is a known trigger of EBV entry into the lytic cycle. It is therefore possible that EBV reactivation (and concomitant EBV-specific T cell expansion) is a marker of B cell activation, which is the true driver of MS pathology. We have modified our discussion to highlight these important considerations (lines 472-476 of the manuscript).

6. "Alternatively, the findings of CD8+ T cells reactive against EBV late latent and lytic antigens are consistent with other reports^{4,41,45} and could indicate EBV reactivation within the CNS."

Comment: EBV reactivation within CNS is not the only explanation of this finding -- peripherally-activated cells could preferentially traffic to the CNS, for example. Further, EBV reactivation is not a stable, unchanging state but one point in a dynamic cycle of lytic and latent infection (see PMID: 39865738 for MS-relevant discussion). The most direct test of this hypothesis, Atara's

phase 2 EMBOLD trial produced a negative result (<https://www.neurologylive.com/view/ata188-fails-meet-primary-end-point-phase-2-embold-study-progressive-ms>) and would need to be discussed in this context.

Response: This reviewer is correct that there are multiple mechanisms by which EBV could gain access to the CNS. Indeed, a number of studies have described the induction of “atypical” T-bet⁺ CXCR3⁺ B cells by EBV^{12–14}, which could be a mechanism by which EBV-infected B cells traffic to the CNS. The reviewer’s point regarding EBV reactivation dynamics is also very important. It is essential to recognize that EBV expression is dynamic as the virus periodically reactivates different latent and lytic programs^{11,15,16}. Given the widespread seroprevalence of EBV in the general population, it is plausible that dysregulation of EBV expression is relevant to MS pathology (notwithstanding the previously cited possibility of an epiphenomenon). As the reviewer has pointed out, there have been several clinical trials using adoptive T cell therapies targeting EBV in MS. While an early phase I trial showed some modest clinical benefit in progressive MS¹⁷, the more recent phase II EMBOLD trial (ATA188) did not show a clear benefit¹⁸. Notably, these studies had a number of key limitations, including focus on progressive MS and reliance on confirmed disability improvement as a primary outcome. Numerous questions remain, including whether EBV-specific T cell therapies alter EBV viral loads and expression as well as relapse and MRI outcomes. We have included these important considerations in the discussion (lines 464-472).

7. "The detection of EBV DNA in the CSF of most MS patients, including those with confirmed expanded EBV351 specific CD8+ T cells, is strongly supportive of this possibility."

Comment: This statement seems overly forceful, given the equivalent frequency of detecting EBV DNA in HC and MS/CIS. Would it be appropriate to assay for CSF DNA from other Herpesviruses including CMV and HHV6?

Response: As described in our response to Reviewer 1, we have now provided ddPCR analysis of EBV transcripts from cDNA corresponding to different stages of EBV latency and lytic infection. *EBER2* cDNA was overall less detectable compared to DNA with no difference between CIS/MS and HC/OND (Fig. 7A-B). However, we observed a significant increase in *BamHI-W* transcripts in CIS/MS patients compared to HC/OND (Fig. 7B). These provides stronger evidence that EBV reactivation is enhanced in CIS/MS, which could drive the expansion of EBV-specific CD8+ T cells. The corresponding new text is found in lines 379-387 of the manuscript.

While comparisons to other herpesviruses, such as CMV and HHV6, could be of further interest, we have focused our analysis on EBV since only EBV-reactive clonally expanded CD8+ T cells were observed in our study.

Figure 7. Detection of EBV DNA and transcripts in CSF. Summary of EBV DNA ddPCR results from CSF supernatant in which *EBER2* was normalized to a housekeeping gene (A). EBV cDNA for each the indicated genes were measured by ddPCR and normalized to a housekeeping gene (B). Each sample was run in duplicate, and each dot represents the average result from each subject (mean and SEM shown).

"The detection EBV in the CSF of non-MS individuals likely reflects the fact that the majority of the general population, with or without MS, is chronically infected with EBV."

Comment: It would be important to say more about this point: HC do not have B cell rich meningeal aggregates, while many MS patients do have these (they are the source of OCBs), and would strongly bias towards finding EBV DNA in the CSF of MS/CIS. Therefore, unless there is an alternative hypothesis, finding EBV DNA in the CSF of both MS/CIS and HC is likely to be an incidental finding. The trafficking B cell population in HC CSF will be vanishingly small, and could not account for the presence of EBV DNA in CSF. The data are consistent with the possibility that cell-free EBV DNA enters the CSF from blood of both MS research participants and healthy controls.

Response: The reviewer raises an important point regarding the potential sources of EBV in the CSF of CIS/MS and HC/OND subjects. The reviewer is absolutely correct that there are only trace frequencies of B cells in healthy CSF whereas CIS/MS patients have significantly higher B cells in the CSF and CNS. We agree that is possible that cell-free EBV DNA in the CSF could be derived from the blood. It is also important to recognize that there are alternate cellular reservoirs of EBV, including in the CNS, which has been reported in healthy individuals. We have added these considerations to the discussion (lines 490-493).

"Pseudotime analysis revealed distinct populations of T cells largely segregated based on compartment (i.e. CSF or blood)"

Is it possible to apply pseudotime analysis across two such different samples as blood and CSF? This approach is typically applied to subpopulations of a single population of cells.

Response: While our goal was to compare transcriptomic trajectories of T cells between blood and CSF (since there is communication between the two compartments), we agree with the reviewer that pseudotime analyses are usually applied to cells within a single tissue. We have therefore removed the prior CSF/blood pseudotime plot and included pseudotime analyses on T cells solely within the CSF (**Fig. 1D**) The corresponding new text is found in lines 110-113 of the manuscript.

D

Figure 1. T cell single cell sequencing analysis in blood and CSF. Pseudotime trajectory analysis of CD4+ (blue) and CD8+ T cells (green) in CSF and PB is shown in D.

10. "153 More than 70% of the highly expanded and CSF-enriched T cell clonotypes in the CSF were CD8+ T cells."

This observation is impressive given that CD4+ cells in CSF outnumber CD8+ cells more than 2:1 and that the authors sequenced twice as many CD4s as CD8s. This bias towards CD8+ cell clonality deserves further comment.

Response: We agree this is a striking finding, and we have inserted additional language into the discussion highlighting the significance of CSF-expanded and enriched CD8+ T cells. The corresponding new text is found in lines 403-405 of the manuscript.

Literature Cited

1. Ban, M. *et al.* Expression profiling of cerebrospinal fluid identifies dysregulated antiviral mechanisms in multiple sclerosis. *Brain* **147**, 554–565 (2024).
2. Ramesh, A. *et al.* A pathogenic and clonally expanded B cell transcriptome in active multiple sclerosis. *Proceedings of the National Academy of Sciences* **117**, 22932 LP – 22943 (2020).
3. Lossius, A. *et al.* High-throughput sequencing of TCR repertoires in multiple sclerosis reveals intrathecal enrichment of EBV-reactive CD8+ T cells. *European Journal of Immunology* **44**, 3439–3452 (2014).
4. Erdur, H. *et al.* EBNA1 antigen-specific CD8+ T cells in cerebrospinal fluid of patients with multiple sclerosis. *Journal of Neuroimmunology* **294**, 14–17 (2016).
5. Gottlieb, A., Pham, H. P. T., Saltarelly, J. G. & Lindsey, J. W. Expanded T lymphocytes in the cerebrospinal fluid of multiple sclerosis patients are specific for Epstein-Barr-virus-infected B cells. *Proceedings of the National Academy of Sciences* **121**, e2315857121 (2024).
6. He, R. *et al.* Follicular CXCR5-expressing CD8+ T cells curtail chronic viral infection. *Nature* **537**, 412–416 (2016).
7. Schmidt, F. *et al.* In-depth analysis of human virus-specific CD8+ T cells delineates unique phenotypic signatures for T cell specificity prediction. *Cell Reports* **42**, 113250 (2023).
8. Glanville, J. *et al.* Identifying specificity groups in the T cell receptor repertoire. *Nature* **547**, 94–98 (2017).
9. Huang, H., Wang, C., Rubelt, F., Scriba, T. J. & Davis, M. M. Analyzing the Mycobacterium tuberculosis immune response by T-cell receptor clustering with GLIPH2 and genome-wide antigen screening. *Nature Biotechnology* **38**, 1194–1202 (2020).
10. Lammoglia Cobo, M. F. *et al.* Rapid single-cell identification of Epstein–Barr virus-specific T-cell receptors for cellular therapy. *Cytotherapy* **24**, 818–826 (2022).
11. Wahbeh, F. & Sabatino, J. J. Epstein-Barr Virus in Multiple Sclerosis. *Neurology Neuroimmunology & Neuroinflammation* **12**, e200460.
12. van Langelaar, J. *et al.* The association of Epstein-Barr virus infection with CXCR3+ B-cell development in multiple sclerosis: impact of immunotherapies. *European Journal of Immunology* **51**, 626–633 (2021).
13. SoRelle, E. D. *et al.* Early multiple sclerosis activity associated with TBX21+CD21^{lo}CXCR3+ B cell expansion resembling EBV-induced phenotypes. *JCI Insight* (2025) doi:10.1172/jci.insight.188543.
14. Jelcic, I. *et al.* T-bet+ CXCR3+ B cells drive hyperreactive B-T cell interactions in multiple sclerosis. *Cell Reports Medicine* **6**, 102027 (2025).
15. Thorley-Lawson D A & Gross A. Persistence of the Epstein–Barr Virus and the Origins of Associated Lymphomas. *New England Journal of Medicine* **350**, 1328–1337 (2004).
16. Giovannoni, G. EBV-specific T-cell responses are telling us something important about multiple sclerosis. *Brain* **148**, 692–694 (2025).
17. Pender, M. P. *et al.* Epstein-Barr virus–specific T cell therapy for progressive multiple sclerosis. *JCI Insight* **3**, (2020).
18. Giovannoni, G., Hawkes, C. H., Lechner-Scott, J., Levy, M. & Yeh, E. A. Emboldened or not: The potential fall-out of a failed anti-EBV trial in multiple sclerosis. *Multiple Sclerosis and Related Disorders* **81**, (2024).